# RNA-binding proteins of KHDRBS and IGF2BP families control the oncogenic activity of MLL-AF4

Hiroshi Okuda [1,2] ✉, Ryo Miyamoto[1], Satoshi Takahashi[1,3], Takeshi Kawamura [4], Juri Ichikawa[2], Ibuki Harada[2], Tomohiko Tamura [2,5] & Akihiko Yokoyama [1,6] ✉

Chromosomal translocation generates the *MLL-AF4* fusion gene, which causes acute leukemia of multiple lineages. MLL-AF4 is a strong oncogenic driver that induces leukemia without additional mutations and is the most common cause of pediatric leukemia. However, establishment of a murine disease model via retroviral transduction has been difficult owning to a lack of understanding of its regulatory mechanisms. Here, we show that MLL-AF4 protein is post-transcriptionally regulated by RNA-binding proteins, including those of KHDRBS and IGF2BP families. *MLL-AF4* translation is inhibited by ribosomal stalling, which occurs at regulatory sites containing AU-rich sequences recognized by KHDRBSs. Synonymous mutations disrupting the association of KHDRBSs result in proper translation of MLL-AF4 and leukemic transformation. Consequently, the synonymous MLL-AF4 mutant induces leukemia in vivo. Our results reveal that post-transcriptional regulation critically controls the oncogenic activity of MLL-AF4; these findings might be valuable in developing novel therapies via modulation of the activity of RNA-binding proteins.

Chromosomal translocations generate oncogenic fusion genes, and the products encoded by such genes drive leukemogenesis of specific lineages depending on the nature of the oncoproteins[1]. Gene rearrangements of Mixed Lineage Leukemia (*MLL*; also known as *KMT2A*) gene account for 7 and 6% of acute myeloid leukemia (AML) and acute lymphoblastic leukemia (ALL) cases, respectively[2]. To date, over 80 fusion partners of MLL have been identified. Nonetheless, approximately 75% cases are caused by fusions with a component of a transcriptional co-activator complex termed AEP (AF4 family/ENL family/P-TEFb)[2,3]. AEP consists of AF4 family proteins, such as AF4 (also known as AFF1) and AF5Q31 (also known as AFF4); ENL family proteins, such as ENL (also known as MLLT1) and AF9 (also known as MLLT3); and the positive transcription elongation factor b (P-TEFb)[3]. A similar complex has been characterized as a transcription elongation factor by others and named as the super elongation complex[4–6]. AEP activates gene expression at multiple steps of transcription. It initiates transcription by loading TATA box binding proteins (TBP) via selectivity factor 1 (SL1)[7,8] and activates transcription elongation by phosphorylating RNA polymerase II (RNAP2) via P-TEFb[3–6,9]. Leukemic MLL-AEP fusion proteins function as constitutively active transcriptional machinery that aberrantly activates a set of oncogenic genes, including *HOXA9* and *MEIS1*, which are typically expressed in hematopoietic stem cells[10,11].

*AF4* is the most frequently fused partner gene of *MLL*, which accounts for one-third of the entire *MLL*-rearranged cases of leukemia.

[1]Tsuruoka Metabolomics Laboratory, National Cancer Center, Tsuruoka, Yamagata, Japan. [2]Department of Immunology, Yokohama City University Graduate School of Medicine, Yokohama, Kanagawa, Japan. [3]Department of Hematology and Oncology, Kyoto University Graduate School of Medicine, Kyoto, Kyoto, Japan. [4]Research Center for Advanced Science and Technology (RCAST), The University of Tokyo, Bunkyo, Tokyo, Japan. [5]Advanced Medical Research Center, Yokohama City University, Yokohama, Kanagawa, Japan. [6]National Cancer Center Research Institute, Chuo, Tokyo, Japan. ✉e-mail: okuda.hir.tv@yokohama-cu.ac.jp; ayokoyam@ncc-tmc.jp

Its occurrence in leukemia is strongly biased to the B cell lineage for unknown reasons. B-ALL (B-lymphoblastic leukemia) cases with MLL-AF4 translocation are the major cause of infant leukemia and often associated with poor prognosis[12]. Animal models that recapitulate the human disease are valuable tools for understanding molecular mechanisms of the disease and developing therapeutic strategies. Studies have shown that MLL fusion genes immortalize murine hematopoietic stem/progenitor cells (HSPCs) ex vivo and induce overt AML in syngeneic mice in vivo[10,11,13–15]. Nevertheless, establishing a mouse disease model that can faithfully recapitulate MLL-AF4-mediated leukemia has been unsuccessful[12,16]. The fusion gene of human *MLL* and *AF4* (*MLL-AF4*) does not exhibit a potent oncogenic activity in experimental models[3,17–19]. A breakthrough came from the groups of Thirman and Mulloy, who demonstrated that the fusion gene of human *MLL* and murine *Af4* (*MLL-mAf4*) caused AML in mouse models[17,20]. Furthermore, *MLL-mAf4* induced pro-B ALL when transduced in human HSPCs and transplanted in immuno-compromised mice. However, the mechanism underlying the inability of human MLL-AF4 to induce leukemia remains unclear, and animal models of MLL-AF4-mediated leukemia in immuno-competent mice are still lacking.

We reasoned that the inability of human MLL-AF4 to induce leukemia may be due to the suppression of its activity by cell-intrinsic regulatory mechanisms. In this work, we dissect the regulatory mechanism of MLL-AF4 and reveal that RNA-binding proteins (RBPs) have the potential to specifically inactivate this oncogene.

## Results

### *MLL-AF4* translation is suppressed post-transcriptionally

The oncogenic activity of MLL fusion proteins can be evaluated using an ex vivo myeloid progenitor transformation assay, in which murine c-Kit positive HSPCs are retrovirally transduced with leukemic *MLL* fusion genes and cultured in semi-solid media in the presence of myeloid cytokines[15,21]. MLL-ENL- and MLL-AF10-transformed murine HSPCs demonstrated vigorous colony forming activity and sustained expression of *Hoxa9*, whereas the MLL fragment lacking the fusion partner portion (MLL ΔFP) and the empty vector produced no colonies in the late passages as reported previously[22,23]. The retrovirus carrying human *MLL-AF4* was unable to immortalize murine HSPCs (Fig. 1a), as reported previously[3]. In contrast, MLL-mAf4, a fusion of human *MLL* and murine *Af4*, potently transformed murine HSPCs (Fig. 1a), consistent with previous reports[17,20].

To assess the oncogenic activity in vivo, we next performed in vivo leukemogenesis assay[15,24], in which various *MLL-AF4* fusions are transduced to murine HSPCs and transplanted into syngeneic mice. MLL-mAf4 induced leukemia in vivo in the recipient mice, whereas MLL-AF4 did not (Fig. 1b). The presence of leukemia-initiating cells was confirmed by secondary transplantation of mice with MLL-mAf4-mediated leukemia cells. To dissect the mechanism underlying the dysfunction of MLL-AF4, we first determined the expression of *MLL-AF4* transcripts after transfection of the MLL-AF4 plasmid into 293 T cells. All the mRNAs of MLL-fusions including *MLL-AF4* were expressed at comparative levels (Fig. 1c), and only the MLL-AF4 protein was not detected in the lysate (Fig. 1d). We next examined the efficiencies of virus production and genome integration, and expression of viral RNA/protein in murine HSPCs. The efficiencies of virus production and genome integration of the virus carrying *MLL-AF4* were not substantially lower than those of the others (Fig. 1e, f and Supplementary Fig. 1a). Nonetheless, the number of *MLL-AF4*-transduced HSPCs after G418 selection was substantially reduced (Fig. 1g), suggesting that some steps in the viral life cycle between integration and expression of the viral protein were impaired. The MLL-AF4 protein was not detected in the HSPCs, whereas the mRNA expression of *MLL-AF4* was detected, albeit at low levels (Fig. 1h, i), indicating that *MLL-AF4* translation was inhibited post-transcriptionally. Finally, we used a retrovirus pool of MLL-AF4 with a titer approximately 60-fold of that of MLL-mAf4, but did not observe

any colony formation in late passages, whereas MLL-mAf4 produced colonies (Supplementary Fig. 1b). When *MLL-AF4* mRNA was transiently expressed from the CMV promoter in 293 T cells, the MLL-AF4 protein was detected (Supplementary Fig. 1c), indicating this inhibitory mechanism of MLL-AF4 can be overcome by substantially increasing the mRNA level. These results suggested that post-transcriptional regulation was the major determinant of the transforming ability of MLL-AF4, but not virus production and its transduction.

### AF4 RNA sequence is responsible for post-transcriptional inactivation of MLL-AF4

To determine the RNA sequence responsible for the post-transcriptional regulation of MLL-AF4, we constructed a series of domain swapping mutants of MLL-AF4 and MLL-mAf4, and examined their protein expression in 293 T cells (Supplementary Fig. 2a, b). Domain swapping analysis revealed a 210 bp sequence (the coding sequence of AF4 corresponding to the sequence 3124/3333) located between the S5 and S1 sites (Fig. 2a), which we identified as the sequence responsible for the post-transcriptional inactivation of MLL-AF4. The transforming ability of these mutants correlated with their protein expression (Supplementary Fig. 2c). Disruption of this regulatory sequence by swapping at the S0 site (Supplementary Fig. 2a), which is located in the middle of the S5−S1 region (Fig. 2a), resulted in the translation of both *MLL-AF4/mAF4* and *-mAf4/AF4* mutants and transformation of murine HSPCs [Supplementary Fig. 2a−c, see MLL-AF4/mAf4 mutants (S0-end)]. Furthermore, exchange of the S5−S1 region of human AF4 and murine mAf4 completely reversed the transforming ability and protein expression (Fig. 2a−c), indicating that the S5−S1 region was necessary and sufficient to induce the post-transcriptional inactivation of MLL-AF4. To examine whether the sequence in the MLL portion influences the protein expression of MLL-AF4, we constructed domain swap mutants of murine/human MLL tethered to murine/human AF4 and examined their oncogenic activities using an ex vivo myeloid progenitor transformation assay (Supplementary Fig. 2d). These results indicate that the human AF4 portion, but not the MLL portion, is critically associated with the post-transcriptional regulation of MLL-AF4. Hence, we defined the S5−S1 region as the post-transcriptional regulatory sequence (PTRS) of human *AF4* (Fig. 2a).

To determine whether this post-transcriptional regulation occurs at the RNA or protein level, we next constructed an MLL-AF4 mutant harboring synonymous mutations at its PTRS (MLL-AF4 sPTRS) and examined its virus titer, protein expression in 293 T cells, and transforming ability of murine HSPCs (Fig. 2a–f). The sPTRS mutations did not affect the efficiency of virus production and genome integration (Fig. 2d, e), and the number of transduced cells after G418 selection was recovered to a normal level comparable to that of *MLL-mAf4* (Fig. 2f), indicating that the PTRS of *AF4* was responsible for inhibiting *MLL-AF4* translation from the viral genome. Translation of *MLL-AF4 sPTRS* in 293 T cells and its transforming ability of murine HSPCs were comparable to those of *MLL-mAf4* (Fig. 2b, c). Because MLL-AF4 causes ALL in human patients, we next evaluated the transforming ability of various MLL-AF4 constructs in a lymphoid condition[17]. Murine HSPCs transduced with *MLL-AF4 sPTRS* or *MLL-mAf4* enhanced proliferative capacities in the lymphoid condition (Fig. 2g). We further constructed a series of MLL-AF4 expression vectors harboring more detailed synonymous mutations within the PTRS and assessed their protein expression and transforming ability (Fig. 2h, i). We identified the S8-S7 region (54 bp) as the minimum PTRS (PTRSmin: the coding sequence of AF4 corresponding to the sequence 3193/3246) of human *AF4* responsible for post-transcriptional regulation. In human leukemia, t(4;11) chromosomal translocation often generates an AF4-MLL gene tethered in-frame, which is suggested to promote leukemogenesis[18]. To examine whether AF4-MLL modulates the post-transcriptional regulation of MLL-AF4, we evaluated the oncogenic effects of AF4-MLL

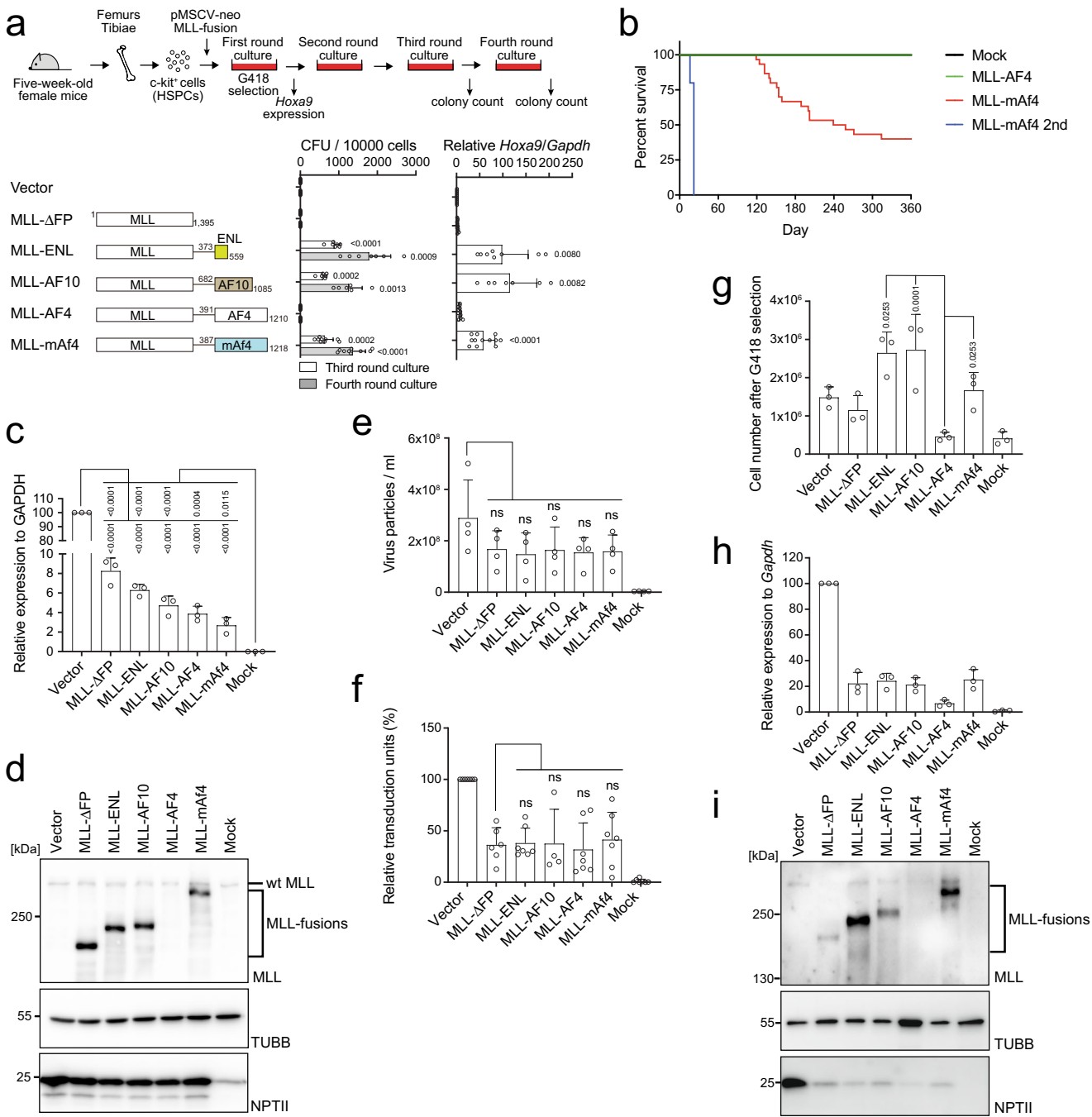

**Fig. 1 | *MLL-AF4* translation is inhibited by post-transcriptional regulation.**
**a** Colony-forming units (CFUs) of various MLL fusions per 10,000 cells at third- and fourth-round passages under myeloid conditions ($n = 9$: Vector, MLL-ΔFP, MLL-AF4, MLL-mAf4; $n = 7$: MLL-ENL; $n = 6$: MLL-AF10). *Hoxa9* expression normalized to that of *Gapdh* of first round colonies is shown as the relative value of the vector control (set to 1) ($n = 12$, Vector, MLL-ΔFP, MLL-AF4, MLL-mAf4; $n = 8$: MLL-ENL; $n = 7$: MLL-AF10). **b** Leukemogenic potential of MLL-AF4 and MLL-mAf4 in vivo. Murine HSPCs were transduced with MLL-AF4 constructs and transplanted into syngeneic mice (mock, $n = 6$; MLL-AF4, $n = 10$; MLL-mAf4, $n = 30$; MLL-mAf4 secondary transplantation $n = 5$). **c** mRNA expression of the *MLL* fusion genes in 293 T cells analyzed using a qPCR probe for the EPS region in the pMSCV neo plasmids ($n = 3$). **d** Western blotting of MLL fusions in 293 T cells transfected with the MLL fusion expression vectors in **c**. Anti-beta tubulin (TUBB) (an internal standard) and neomycin phosphatase II (NPTII) antibodies (a gene transduction control) were included for comparison. **e** Virus particle production by the MLL

fusion expression vectors was quantitated using qRT-PCR with the MSCV-EPS probe and absolute quantification methods ($n = 4$). **f** Transduction of recombinant viruses carrying various *MLL* fusion genes in murine HSPCs were determined using qPCR probes for MSCV-EPS and the murine *Gapdh* locus ($n = 7$: Vector, MLL-ENL, MLL-AF4, MLL-mAf4, Mock; $n = 6$: MLL-ΔFP; $n = 4$: MLL-AF10). **g** Cell numbers of murine HSPCs infected with various retroviruses carrying *MLL* fusion genes after 5 days of selection with G418 ($n = 3$). **h** Relative mRNA expression of *MLL* fusion genes after antibiotic selection using qRT-PCR with the MSCV-EPS probe ($n = 3$). **i** Western blotting of HSPCs transduced with retroviruses carrying *MLL* fusion genes cultured in G418 for 5 days, as shown in **d**. Data are presented as the mean ± SD of biologically independent replicates (**a, c, e, f, g, h**). *P*-value was calculated by one-way ANOVA followed by Tukey's test (**a, c, e, f, g**). Western blotting was performed on two biological replicates (**d, i**). See also Supplementary Fig. 1. Source data are provided as a Source Data file.

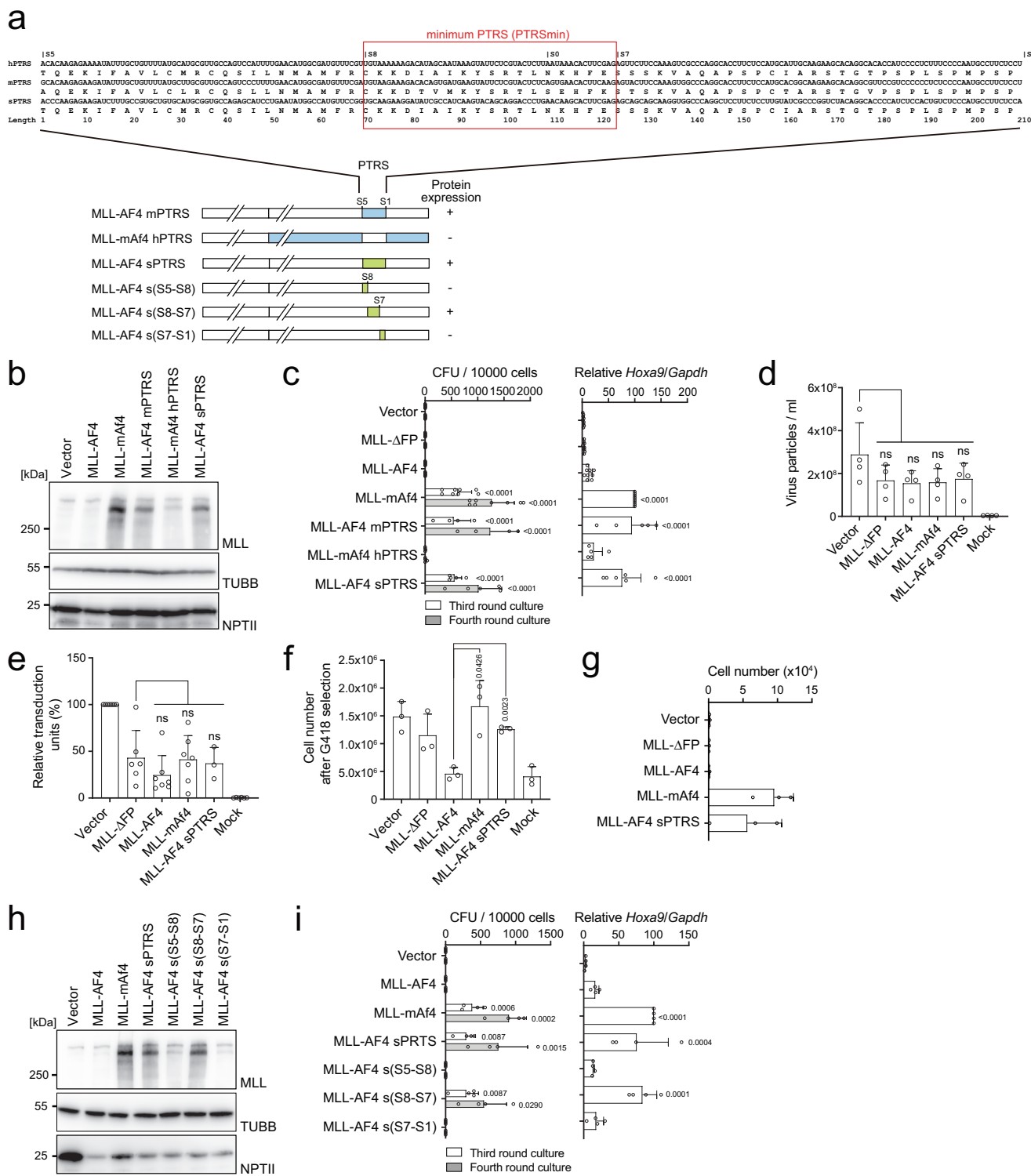

on MLL-AF4-mediated transformation by ex vivo myeloid progenitor transformation assays. AF4-MLL did not exhibit transforming property by itself or in combination with MLL-AF4 in this setting (Supplementary Fig. 2e–g). These results unequivocally demonstrated that the translation of *MLL-AF4* was inhibited at the RNA level.

### A *synonymous* mutant of MLL-AF4 induces leukemia in immunocompetent mice

To investigate the leukemogenic potential of MLL-AF4 sPTRS in vivo, murine HSPCs were transduced with *MLL-AF4 sPTRS* and transplanted into syngeneic mice. MLL-AF4 sPTRS induced leukemia with

similar latency but lower penetrance than MLL-mAf4 (Fig. 3a). Previous attempts to develop murine MLL-AF4 leukemia models using knock-in strategies have yielded mixed outcomes. Chen et al. and Metzler et al. developed constitutive and conditional knock-in mouse lines, respectively, for the fusion gene of mouse *Mll* and human *AF4* (*mMll-AF4*), which induced B-cell lymphoma predominantly[25,26]. In contrast, Krivtsov et al. demonstrated that the conditional expression of mMll-AF4 induced a spectrum of diseases ranging from AML to pre-B ALL with a few cases of pro-B ALL and mixed-phenotype acute leukemias (MPAL)[27]. Although the mechanisms underlying these differences are unclear, recapitulating the human pro-B ALL

**Fig. 2 | The post-transcriptional regulatory sequence is responsible for *MLL-AF4* translation. a** Domain swapping mutants of various MLL-AF4 mutants are shown in blue (mouse), white (human), and green (synonymous mutations). RNA and amino acid sequences of the post-transcriptional regulatory sequence (PTRS) of human AF4 (hPTRS) and its corresponding sequences of mouse AF4 (mPTRS) and the synonymous mutant (sPTRS). The minimum PTRS is indicated with a red rectangle. **b** Western blotting of MLL-AF4 mutants in 293 T cells, as described in Fig. 1d. **c** Transforming ability of MLL-AF4 mutants under myeloid conditions, as described in Fig. 1a ($n = 8$: Vector, MLL-ΔFP, MLL-AF4, MLL-mAf4; $n = 5$: MLL-AF4 sPTRS; $n = 4$: MLL-AF4 mPTRS, MLL-mAf4 hPTRS). *Hoxa9* expression normalized to that of *Gapdh* of first round colonies is shown as the relative value of the MLL-mAf4 (set to 100) ($n = 9$, Vector, MLL-AF4, MLL-mAf4; $n = 8$: MLL-ΔFP; $n = 6$: MLL-AF4 sPTRS; $n = 5$: MLL-AF4 mPTRS, MLL-mAf4 hPTRS). **d** Virus particle production by the MLL-AF4 mutant expression vectors. Virus particle production was quantitated as described in Fig. 1e ($n = 4$). **e** Relative transduction units of retroviruses carrying various *MLL-AF4* mutant genes in murine HSPCs were determined as described in Fig. 1f ($n = 7$, Vector, MLL-AF4, MLL-mAf4; $n = 6$: MLL-ΔFP, Mock; $n = 3$: MLL-AF4 sPTRS). **f** Cell numbers of *MLL-AF4*-transduced murine HSPCs after 5 days of G418 selection ($n = 3$). **g** Transforming ability of MLL-AF4 mutants under an ex vivo lymphoid culture condition. Cell numbers per 10,000 cells at the third passage is shown ($n = 3$). **h** Western blotting of the various MLL-AF4 synonymous mutants in 293 T cells, as described in Fig. 1d. **i** Transforming ability of various MLL-AF4 synonymous mutants under myeloid conditions. CFUs per 10,000 cells in third- and fourth-round culture is shown ($n = 4$). *Hoxa9* expression normalized to *Gapdh* is shown as the relative value of MLL-mAf4 (set to 100) ($n = 4$). Data are presented as the mean ± SD of indicated biologically independent replicates (**c, d, e, f, g, i** P-value was calculated by one-way ANOVA followed by Tukey's test (**c, d, e, f, i**). Western blotting was performed on two biological replicates (**b, h**). See also Supplementary Fig. 2. Source data are provided as a Source Data file.

in mouse models has been difficult. Our retroviral transduction experiments of *MLL-mAf4* into murine HSPCs resulted in AML in vivo in mouse transplantation models (Fig. 3a and Supplementary Fig. 3a), which agreed with the results of a previous report[17]. Given that transduction of the same *MLL-mAf4* gene into human HSPCs and transplantation in immune-compromised mice induced pro-B ALL[17], it appears that murine HSPCs tend to develop myeloid lineage leukemia through the MLL-AF4 fusion protein, whereas human HSPCs tend to develop lymphoid lineage leukemia. In agreement with this assumption, MLL-AF4 sPTRS induced myeloid leukemia in the B220⁻ CD3e⁻ CD11b⁺ immunophenotype (Fig. 3a, b). These leukemic cells induced secondary leukemia with a shorter latency (Fig. 3a). The leukemia cells were CD11b/Gr1-double positive (Fig. 3c), and expressed posterior *Hox* genes, including *Hoxa6*, *Hoxa7*, *Hoxa9*, *Hoxa10, and Hoxa11*, and several other genes including *Meis1*, *c-Kit*, and *Myc*, which were commonly observed in AML cells induced by MLL-mAf4 and other MLL fusions (i.e., MLL-ENL) (Fig. 3d)[10,11,17]. RNA-sequencing analysis demonstrated that MLL-AF4 sPTRS leukemia cells had an expression profile more similar to MLL-mAf4 leukemia cells than to MLL-ENL leukemia cells (Supplementary Fig. 3b). These results demonstrated that AML can be generated in immuno-competent mice by a synonymous mutant of MLL-AF4 and suggested that post-transcriptional regulation plays a critical role in controlling the oncogenic activity of MLL-AF4.

## RNA-*binding* proteins specifically associate with the PTRS of human *AF4*

To identify the regulatory factors responsible for the post-transcriptional regulation of MLL-AF4, we next purified proteins associated with the PTRS of human *AF4*. We performed pull-down assays using a series of synthesized biotinylated RNA oligonucleotides of the PTRSmin of human *AF4* (denoted as hPTRSmin) and streptavidin-conjugated magnetic beads. Various RNA RBPs were specifically co-precipitated with hPTRSmin from the lysate of 293 T cells, but not with the corresponding mouse RNA sequence (denoted as mPTRSmin) or its synonymous mutant (denoted as sPTRSmin) (Fig. 4a). Mass spectrometry analysis identified heterogeneous nuclear ribonucleoproteins (hnRNPs), insulin growth factor 2 binding proteins (IGF2BPs)1/2/3, and KH domain-containing, RNA-binding, signal transduction-associated proteins (KHDRBSs)1/3 as the PTRS-associated factors (Fig. 4b). RNA pull-down assays followed by western blot analysis confirmed that many RBPs (e.g., KHDRBS1/3, ELAVL1) specifically bound to hPTRSmin (Fig. 4c and Supplementary Fig. 4a), whereas IGF2BPs also associated with mPTRSmin and sPTRSmin. Next, we performed a computational analysis to identify the consensus recognition sites for miRNAs, lncRNAs, and the associated RBPs[28], and found three AU-rich KHDRBS binding sites (denoted as AU1, AU2, and AU3, respectively) closely located at the center of hPTRSmin (Fig. 4d and Supplementary Fig. 4b, c).

As KHDRBS proteins function as dimers to simultaneously recognize multiple AU-rich sites in close proximity[29,30], we generated a series of mutants harboring simultaneous mutations at multiple AU-rich sites to analyze their associated factors. Myeloid progenitor transformation assay demonstrated that two of the three KHDRBS binding sites were required for the post-transcriptional inactivation of MLL-AF4, with the second AU-rich site (AU2) being the most influential (Fig. 4e). Accordingly, *MLL-AF4* translation was restored by simultaneous mutations of all three KHDRBS binding sites (Fig. 4f, see MLL-AF4 sAU123). RNA-pull down assays showed that KHDRBS1 did not bind individually to each of the three sites on the RNA; however, disruption of all three KHDRBS binding sites completely abrogated its interaction (Fig. 4g and Supplementary Fig. 4d, e). Furthermore, IGF2BP proteins bound to the PTRS irrespective of mutations, although their binding efficiencies varied depending on the mutations. For instance, synonymous mutations in AU3, which overlaps with the IGF2BP binding site, resulted in attenuation of IGF2BP1 binding (Fig. 4g). Replacement of the human AU3 by mouse AU3 did not alter association of IGF2BPs (Supplementary Fig. 4e, see hPTRS mAU3), which is consistent with the observation that mPTRS binds to IGF2BPs. An MLL-mAf4 mutant in which three human KHDRBS binding sites were artificially introduced, showed reduction in protein expression (Supplementary Fig. 4f) and attenuated trans-forming ability (Supplementary Fig. 4g). Sequence alignment analysis of the region corresponding to the PTRS of AF4 in various species indicated that the three AU-rich sites and their proximal sequences were evolutionarily conserved, especially in primates but were divergent in rodents (Supplementary Fig. 4h). Furthermore, the RBP levels varied among leukemia cell lines of different lineages (Supplementary Fig. 4i), suggesting that the combination of available RBPs varied with the context, which presumably affected the post-transcriptional regulation of MLL-AF4. Taken together, these results suggested that the PTRS of human *AF4* forms RBP/RNA complexes of various compositions depending on the availability of AU-rich sites and RBPs.

## IGF2BP3 is responsible for the post-transcriptional inactivation of *MLL-AF4*

To identify the factors responsible for the post-transcriptional regulation of MLL-AF4, we used CRISPR/Cas9-mediated knock-out screening of the genes encoding the RBPs associated with the PTRS of human *AF4*. First, we infected murine HSPCs with a retrovirus carrying *MLL-AF4* and a lentivirus carrying Cas9 and sgRNAs against various RBP genes, and examined their colony forming ability; however, colonies were not formed (Supplementary Fig. 5a). We reasoned that the RBPs may complementarily function to regulate *MLL-AF4* translation and therefore, knock-out of one factor may not be sufficient to restore the protein expression. To circumvent this possibility, we next used the MLL-AF4 sAU13 mutant, which does not

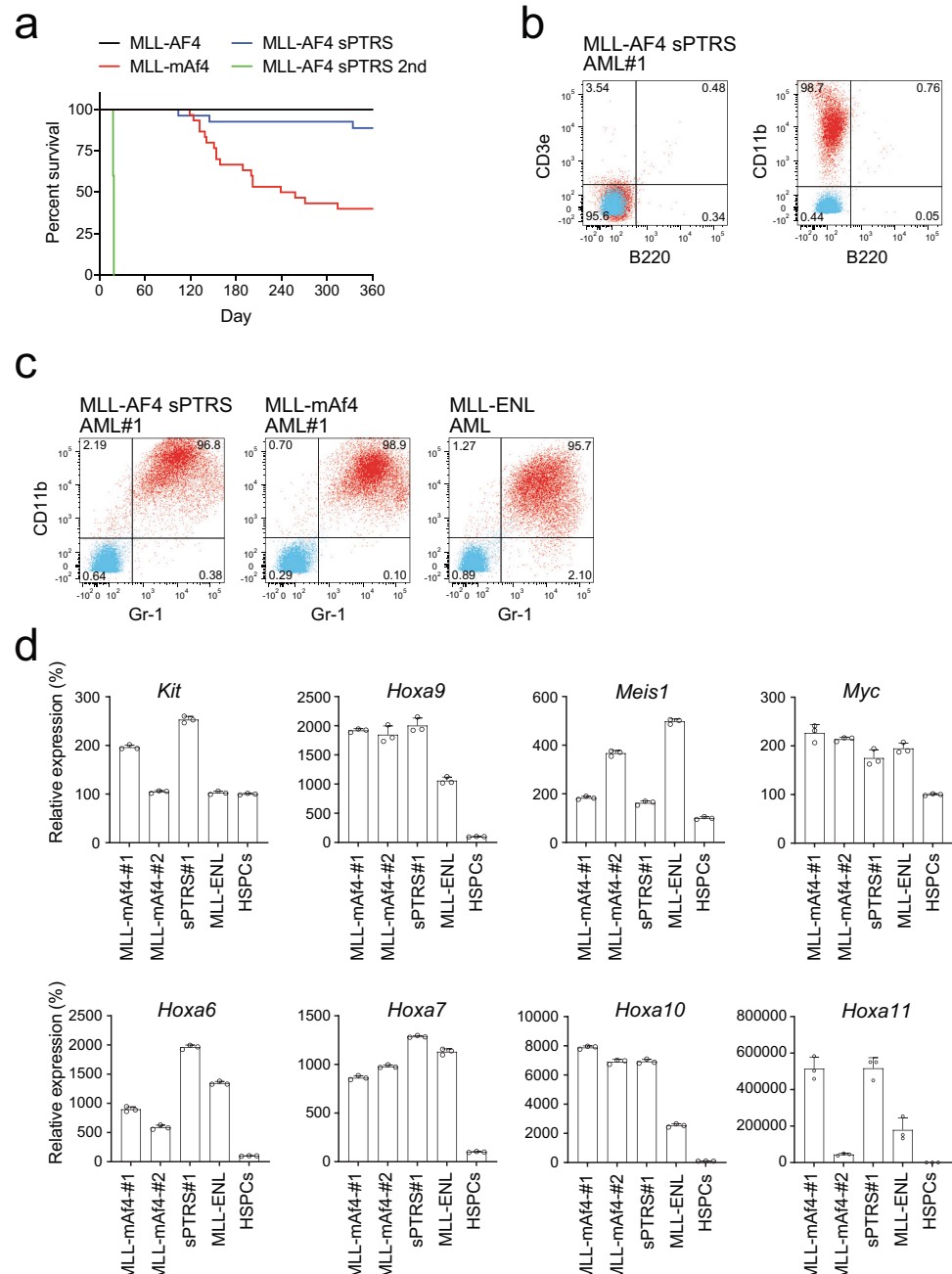

**Fig. 3 | A synonymous mutant of MLL-AF4 induces leukemia. a** Leukemogenic potential of MLL-AF4 sPTRS in vivo. Murine HSPCs were transduced with retrovirus for the MLL-AF4 sPTRS construct and transplanted into syngeneic mice. MLL-AF4 sPTRS, *n* = 30. The B/M-MPAL and T-ALL cases were excluded as retrovirus genome integration was not detected (Supplementary Fig. 3). The MLL-AF4 and MLL-mAf4 data (Fig. 1b) are shown for comparison. **b** The immunophenotype of MLL-AF4 sPTRS-mediated leukemia. Bone marrow cells from mice with MLL-AF4 sPTRS-mediated leukemia were analyzed via flow cytometry for the indicated markers. Unstained control is indicated in blue and stained sample indicated in red. **c** The expression of myeloid markers, Gr1 and Cd11b, in AML cells of MLL-AF4 sPTRS-leukemia. MLL-mAf4- and MLL-ENL-mediated AMLs were included for comparison. **d** qRT-PCR using bone marrow cells from mice with MLL-AF4 sPTRS-mediated leukemia. Expression levels normalized to *Gapdh* are shown relative to those of HSPCs. Data are presented as the mean ± SD of PCR triplicates. See also Supplementary Fig. 3. Source data are provided as a Source Data file.

transform murine HSPCs, but binding of which with KHDRBSs was substantially attenuated (Fig. 4g), expecting it to restore its protein expression relatively easily (Fig. 4f). Indeed, knock-out of *Igf2bp3* enabled transformation of HSPCs by the MLL-AF4 sAU13 mutant (Fig. 5a and Supplementary Fig. 5b). Knock-out of *Hnrnpr*, a gene encoding another RBP, also restored the transforming ability (Supplementary Fig. 5b). Western blot analysis confirmed that *MLL-AF4 sAU13* translation was restored by knocking out of *Igf2bp3* in the immortalized cells (Fig. 5b). In contrast, overexpression of human

IGF2BP3 impaired the proliferation of MLL-AF4 sAU13/sg-*Igf2bp3*-immortalized cells (Fig. 5c). In agreement with this, knock-down of *IGF2BP3* in 293 T cells enhanced the translation of *MLL-AF4 sAU13*, but not that of *MLL-mAF4* or *MLL-AF4* (Supplementary Fig. 5c). Recently, IGF2BP3 has been reported to function as an oncogenic factor in MLL-mAf4 mediated leukemia[31]. Thus, we evaluated the effects of *Igf2bp3* knockout on various MLL fusions in vitro and in vivo (Supplementary Fig. 5d, e). Although we observed a mild decrease in colony formation by MLL-mAf4, we did not detect any substantial

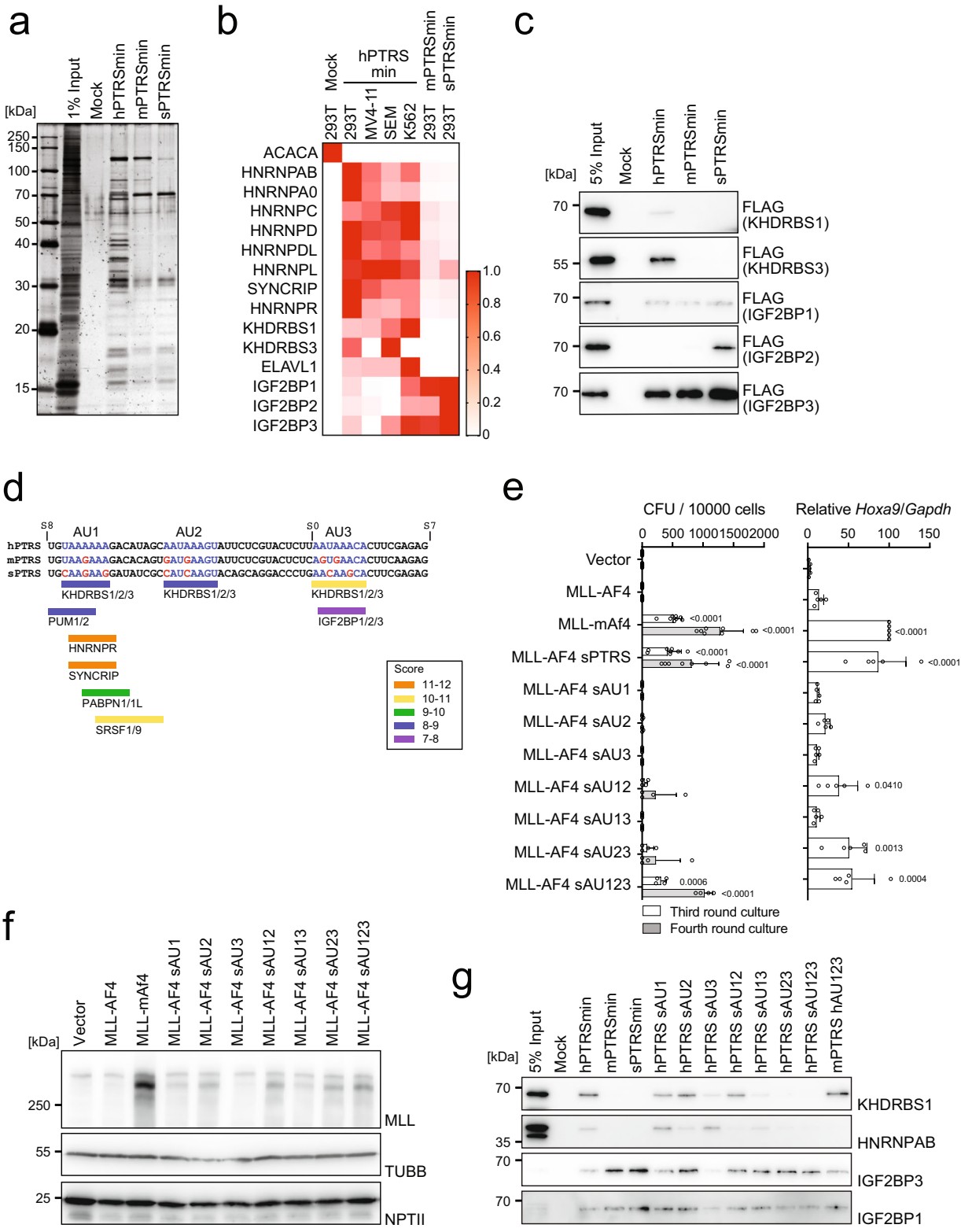

reduction of colony formation induced by mMll-mAf4 or other MLL fusions ex vivo (Supplementary Fig. 5d). Moreover, *Igf2bp3* knockout accelerated the onset of leukemia by MLL-AF10 in vivo in our experimental settings ($p = 0.0377$). These results indicate that *Igf2bp3* does not have strong oncogenic roles in MLL fusion-mediated leukemogenesis in our experimental settings. The MLL-AF4

sAU13/sg-*Igf2bp3*-immortalized cells were less proliferative compared to MLL-mAf4-immortalized cells (Fig. 5a) and could not induce overt leukemia in vivo in this experimental setting (Supplementary Fig. 5e)[32]. These results indicated that IGF2BP3 mediates the post-transcriptional inactivation of MLL-AF4 presumably by collaborating with other RBPs including KHDRBSs and HNRNPR.

**Fig. 4 | Unique RNA-binding proteins specifically bind to RNA containing the PTRS. a** Silver staining of purified factors associated with the minimum PTRS of *AF4*. **b** Heat map of proteins co-purified with the PTRS RNAs with the relative scores. Proteins were identified using mass spectrometry. **c** Immunoprecipitation-western blotting of proteins associated with the PTRS RNAs using 293 T cells transiently expressing FLAG-tagged proteins. **d** CISBP-RNA database analysis of the RNA sequences specifically recognized by various RBPs in the minimum PTRS of human *AF4* (http://cisbp-rna.ccbr.utoronto.ca/index.php). The nucleotide bases in the three AU-rich sites are marked in blue letters, whereas the different bases from the human sequence in the three AU-rich sites are in red. **e** Transforming ability of MLL-AF4 mutants. Various MLL-AF4 constructs carrying synonymous mutations on the AU-rich sites were examined for transformation of HSPCs under an ex vivo myeloid

condition, as shown in Fig. 1A (*n* = 8: Vector, MLL-AF4, MLL-mAf4, MLL-AF4 sPTRS; *n* = 4: the others). *Hoxa9* expression normalized to *Gapdh* is shown as the relative value of MLL-mAf4 (set to 100) (*n* = 5). Data are presented as the mean ± SD of indicated biologically independent replicates. *P*-value was calculated by one-way ANOVA followed by Tukey's test. **f** Western blotting of MLL-AF4 mutants in 293 T cells transfected with the MLL fusion expression vectors shown in (**e**) (as in Fig. 1d). **g** Association of endogenous RBPs with the minimum PTRS of *AF4*. Immunoprecipitation-western blotting was performed using 293 T cells. Endogenous proteins were detected using specific antibodies. Silver staining (**a**) and Western blotting (**c**, **f**, **g**) were performed on two biological replicates. See also Supplementary Fig. 4. Source data are provided as a Source Data file.

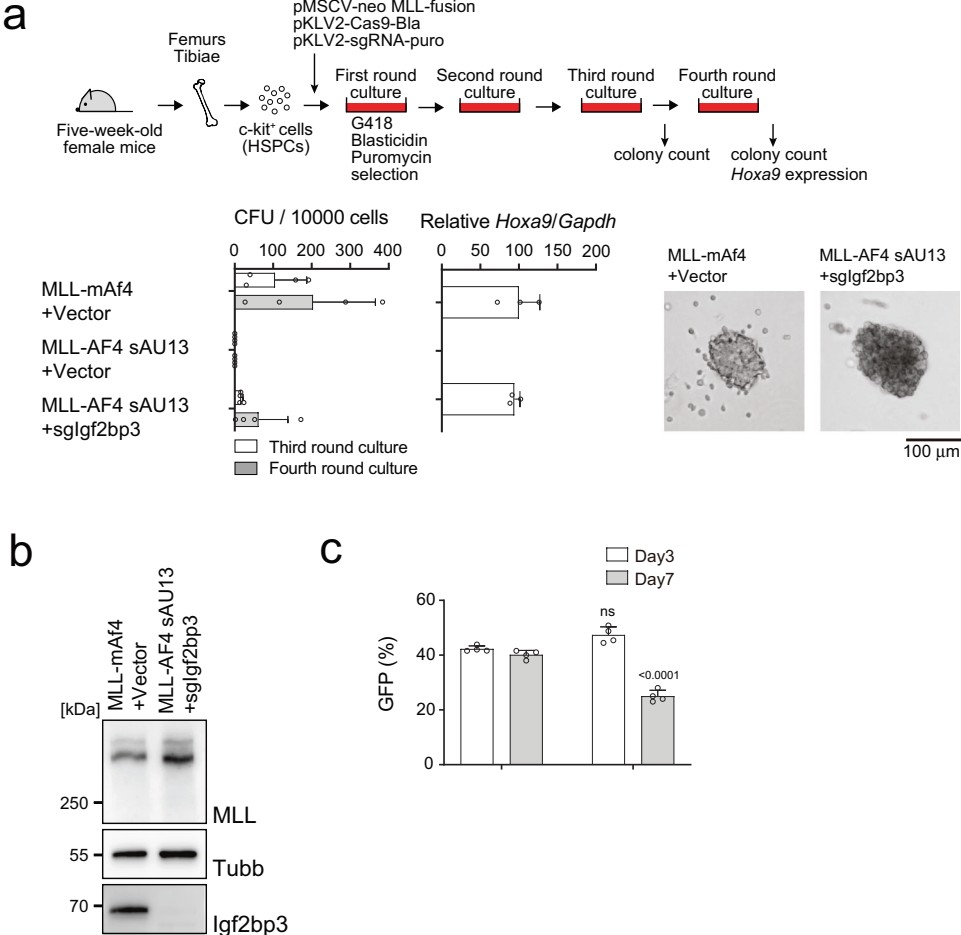

**Fig. 5 | IGF2BP3 is responsible for the post-transcriptional inactivation of MLL-AF4. a** Transforming ability of MLL-AF4 sAU13 in the absence of *Igf2bp3*. The MLL-AF4 construct carrying synonymous mutations at AU1 and 3 was examined for transformation of HSPCs under an ex vivo myeloid condition with co-transduction of a knockout construct for *Igf2bp3*. CFUs are shown as in Fig. 1a, with images of representative colonies (*n* = 4). *Hoxa9* expression during fourth-round passage is shown (*n* = 3). **b** Western blotting of MLL-AF4 sAU13 mutant and endogenous murine IGF2BP3 in immortalized cells. Western blotting was performed on two

biological replicates. **c** Inhibition of MLL-AF4-immortalized cell proliferation by rescuing human IGF2BP3 expression. Cells immortalized by MLL-AF4 sAU13 were transduced with a lentivirus carrying IGF2BP3 and GFP. The ratio of GFP-positive cells was monitored on days 3 and 7 (*n* = 4). Data are presented as the mean ± SD of indicated biologically independent replicates (**a**, **c**). *P*-value was calculated by two tailed *T*-test (**c**). See also Supplementary Fig. 5. Source data are provided as a Source Data file.

## The PTRS of human AF4 induces ribosomal stalling

To gain insights into the mechanism of post-transcriptional inactivation of MLL-AF4, we first established a reporter system to quantitatively evaluate the extent of inactivation using the self-cleaving 2 A sequence sandwiched by HA-tagged RFP and FLAG-tagged GFP in 293 T cells (Fig. 6a). In this system, RFP and GFP were translated as one polypeptide and self-cleaved to produce two fluorescent proteins with one-to-one molecular ratio (Supplementary Fig. 6a). We generated a series of GFP reporters fused with various PTRS derivatives (Fig. 6a). Introduction of

the PTRS of human *AF4* (GFP-hPTRS) selectively attenuated the fluorescence of GFP, whereas its murine counterpart (GFP-mPTRS) and synonymous mutant (GFP-sPTRS) did not (Fig. 6b, c). Reduction of the GFP signals was also observed in the mPTRS reporter carrying three human AU-rich sites (GFP-mPTRS hAU123), whereas the GFP signals were recovered by various combinations of synonymous mutations of human AU-rich sites (Fig. 6b, c). Protein expression of the two fluorescent reporters in western blot analysis showed that the GFP-hPTRS reporter was incompletely translated (Fig. 6d and Supplementary

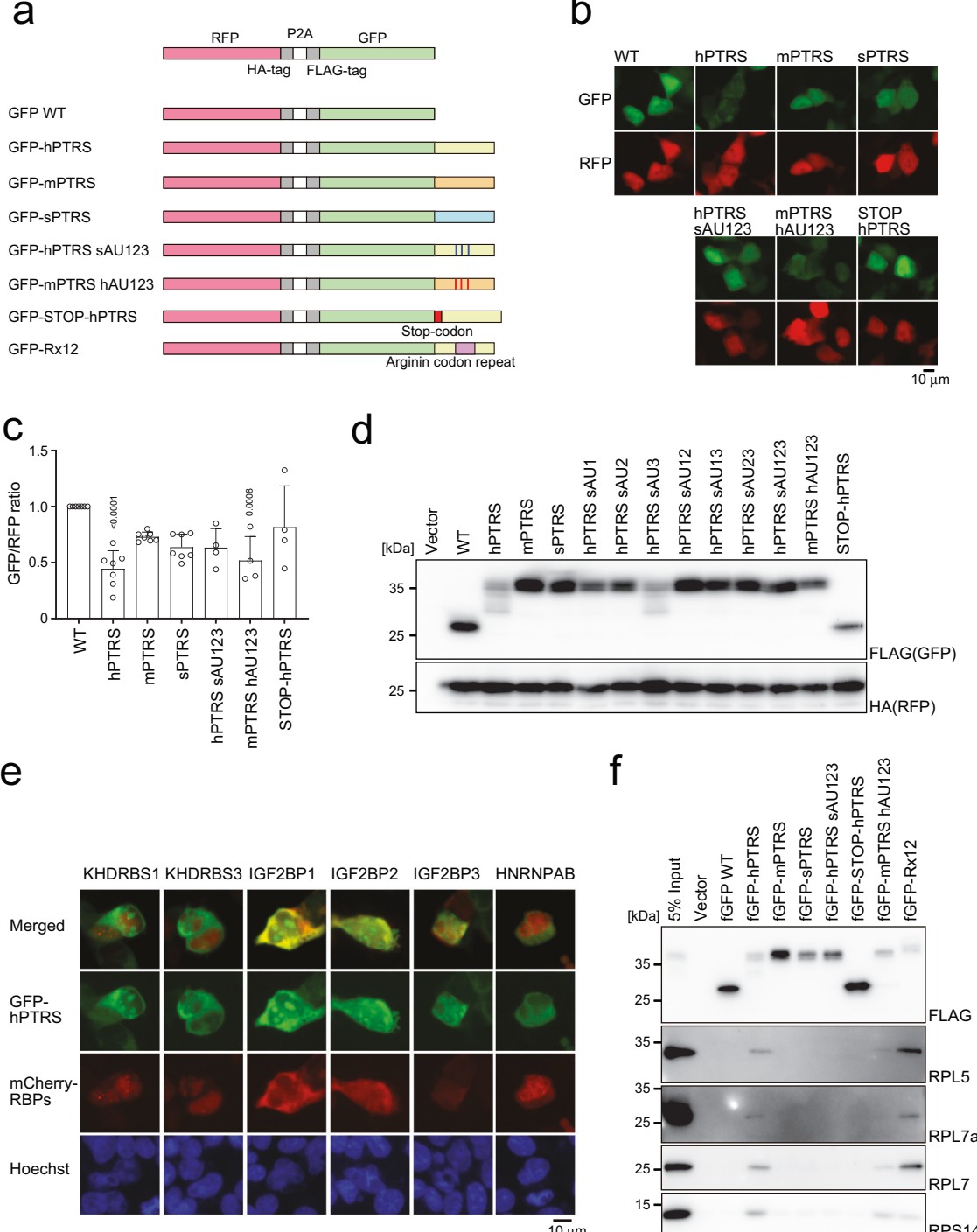

**Fig. 6 | RBP complex inhibits translation of MLL-AF4. a** Schematic representation of the reporter constructs for post-transcriptional regulation. **b** Subcellular localization of GFP and RFP reporters in 293 T cells. **c** Relative fluorescent signals of GFP and RFP reporters. The fluorescent intensities were analyzed using ImageJ software and expressed as the GFP/RFP ratio ($n = 7$: WT, hPTRS, mPTRS, sPTRS; $n = 4$: hPTRS sAU123, mPTRS hAU123, STOP-hPTRS). Data are presented as the mean ± SD of indicated biologically independent replicates. *P*-value was calculated by one-way ANOVA followed by Tukey's test. **d** Western blotting of the GFP/RFP reporter proteins in 293 T cells transiently expressing the constructs. GFP and RFP proteins were detected using FLAG and HA antibodies, respectively. **e** Co-localization of GFP reporter and RNA-binding proteins. The GFP-tagged hPTRS reporter and mCherry-tagged RBPs constructs were co-transfected in 293 T cells. Nuclei were stained with Hoechst 33342. **f** Association of ribosomal proteins with GFP reporters. FLAG-tagged GFP reporters transiently expressed in 293 T cells were analyzed via immunoprecipitation-western blotting with anti-FLAG antibody. Endogenous proteins were detected using specific antibodies. Western blotting was performed on two biological replicates (**d**, **f**). See also Supplementary Fig. 6. Source data are provided as a Source Data file.

Fig. 6b–d). Insertion of a stop codon between the open reading frame of GFP and PTRS of human *AF4* completely blocked the post-transcriptional regulation (Fig. 6a–d), indicating that the post-transcriptional inactivation of AF4 was coupled with translation.

A substantial amount of the fluorescence signal indicated that the GFP-PTRS fusion of human AF4 was present in the cytosol and co-localized with IGF2BPs, but not with KHDRBSs or HNRNPAB (Fig. 6e). A 2-step IP analysis showed that these RBPs were not in the same protein/RNA

complex (Supplementary Fig. 6e), suggesting that IGF2BP3 and KHDRBS3 may independently regulate the amount of AF4 protein in cytosol and nucleus respectively. Mass spectrometry of co-purified proteins with GFP-hPTRS revealed ribosomal proteins (Supplementary Fig. 6f), indicating that many of the nascent GFP-hPTRS reporter proteins were still complexed with ribosomes. These results indicated that translation arrest may be caused by ribosomal stalling at the AU-rich sites on the PTRS, potentially mediated by IGF2BPs. Immunoprecipitation-western blot analysis for the GFP-hPTRS reporter confirmed the association of ribosomal proteins (Fig. 6f). A GFP reporter fused with a polyarginine stalling sequence also co-precipitated ribosomal proteins as a positive control for ribosomal stalling[33]. These results collectively indicated that stalling of translational ribosomes occurred at the AU-rich sites of the PTRS of human *AF4*.

## Discussion

In this study, we showed that the expression of the MLL-AF4 protein is regulated post-transcriptionally by RBPs, including IGF2BPs and KHDRBSs. Translation of *MLL-AF4* is inhibited by ribosomal stalling at the AU-rich sites recognized by KHDRBSs, which are located in the coding region of the human AF4 mRNA. By introducing synonymous mutations that disrupt the binding of KHDRBSs, we generated an MLL-AF4 mutant that can be sufficiently expressed to exert its oncogenic ability. This synonymous MLL-AF4 mutant induced myeloid leukemia in syngeneic mice. Thus, our results demonstrate that post-transcriptional regulation by RBPs controls the efficiency of MLL-AF4 translation, thereby determining its oncogenic activity.

Although MLL-AF4 is the major cause of *MLL*-rearranged leukemia, establishment of a mouse model that recapitulates the human disease has been challenging[16]. Two previous studies have provided the most convincing mouse models for MLL-AF4-mediated leukemia. Krivtsov et al. developed a conditional knock-in allele that expresses mMll-AF4[27]. This model provided a spectrum of diseases ranging from AML to B-ALLs, including pre-B and pro-B ALLs and B/M-MPAL, most of which expressed *Hoxa9* and *Meis1*. Lin et al. also developed a valuable model of MLL-AF4-mediated leukemia wherein *MLL-mAf4* was retrovirally transduced into human HSPCs and transplanted into immuno-compromised mice, which developed pro-B ALL with the CD19+ CD10− CD34+ immunophenotype[17]. The ALL cells were mostly CD19+, and partially CD34+, with an expression profile similar to that of human pro-B leukemia cells. They characteristically expressed *RUNX1* and *MEIS1*, but not the *HOX*-A genes. The same MLL-mAf4 construct developed AML when transduced into mouse HSPCs, which characteristically expressed *Hoxa9* and *Meis1*, suggesting that human HSPCs are prone to developing pro-B ALL compared to mouse HSPCs. In agreement with this, MLL-mAf4 also predominantly developed AML under our experimental conditions. These results support the hypothesis that murine HSPCs are intrinsically prone to developing AMLs and are thus not suitable for modeling ALLs. In this study, an MLL-AF4 synonymous mutant induced myeloid leukemia in murine bone marrow transplantation models. The phenotype of MLL-AF4 sPTRS-mediated AMLs was similar to that of MLL-mAf4- and MLL-ENL-induced disease, as they expressed *Cd11b* and *Gr1* (Fig. 3c). The leukemia cells expressed posterior *Hox* genes, *Meis1*, *c-Kit*, and *Myc* (Fig. 3d). Our results indicated that post-transcriptional regulation of MLL-AF4 plays an important role in controlling its oncogenic activity. We have established an MLL-AF4-mediated murine AML leukemogenesis model via retroviral transduction by modulating post-transcriptional regulation. During the development of leukemia in humans, leukemic cells must somehow circumvent this regulatory mechanism, perhaps by selecting the right cellular context that allows *MLL-AF4* translation. As MLL-AF4-mediated leukemia is most common in B-ALLs in infants, it is intriguing to speculate that such contexts play a decisive role in the selection of lineages and/or developmental stages of the cell of origin. Recently, fetal hematopoietic contexts of the human cord blood cells have shown to confer transformation by MLL-AF4

which led to the development of B-ALL[34,35]. Moreover, miR-130 and miR-128a have been shown to influence on the lineage choice of the MLL-AF4 mediated leukemia[36]. These specific cellular contexts may determine the post-transcriptional regulation of MLL-AF4.

Post-transcriptional regulation is a mechanism for achieving appropriate protein expression by modulating multiple steps, including RNA splicing, polyadenylation, capping, base modification, and localization. RBPs specifically regulate the fate of RNAs by recognizing evolutionarily conserved sequences of RNAs[28]. For example, AU-rich elements (AREs) are one of the determinants of mRNA stability, which recruit specific nuclease complexes to maintain the quality of mRNAs. AREs have a core sequence of AUUUA and are normally found in the 3′ untranslated region (3′ UTR) of mRNAs, where they are recognized by multiple RBPs[37]. We identified three short AU-rich sites in the coding sequence of *AF4*, with which KHDRBSs specifically associate to regulate the RNA processing events, including splicing, export, stability, and translation[29,30]. Furthermore, IGF2BP proteins also regulate RNA stability and translation by forming a higher-order messenger ribonucleoprotein structure in the cytoplasm[38–41]. We demonstrated that a combination of *Igf2bp3* knock-out and synonymous mutations at two of the three AU-rich sites resulted in the restoration of MLL-AF4 protein expression and transformation of HSPCs. These results demonstrated that *MLL-AF4* translation is post-transcriptionally regulated by multiple RBPs, including IGF2BPs and KHDRBSs, and that this regulation impacts the oncogenic activity of MLL-AF4. IGF2BP3 has been reported also as an oncogenic factor implicated in the transcription and splicing of MLL target genes in MLL-mAf4 mediated leukemia as its knockout impaired leukemogenesis in a mouse disease model[31]. In our experimental settings, we did not see a substantial reduction of leukemogenic activity of various MLL fusion-immortalized cells by knocking out Igf2bp3 (Supplementary Fig. 5d, e). IGF2BP3 may function as a bi-directional regulatory factor for MLL-AF4-mediated leukemogenesis, which modulates MLL-AF4 translation in the anti-oncogenic direction and its target gene transcription/splicing in the pro-oncogenic direction.

Our protein localization analysis using GFP reporters showed that KHDRBSs are localized mainly in the nucleus, whereas IGF2BPs are localized in the cytosol. The two-step IP analysis indicated that these factors function independently on the AU-rich motifs. These data suggest that the RBPs regulate MLL-AF4 expression in multiple steps at multiple locations, including translation and RNA export. Ribosomal stalling is a mechanism of controlling protein production[33,42,43]. Our PTRS reporter analysis showed that the three AU-rich sites within the PTRS are involved in ribosomal stalling, as shown by premature halting of translation and association with ribosomal complexes (Fig. 6c and Supplementary Fig. 6e). Abrogation of the AU-rich sites restored the protein expression and promoted cell transformation. Thus, we concluded that ribosomal stalling occurs at the AU-rich sites in the human *AF4* mRNA. Our findings strongly indicate that MLL-AF4 is post-transcriptionally regulated, which explains why the establishment of a mouse disease model by MLL-AF4 has been difficult. However, our study does not directly address why MLL-AF4 efficiently causes leukemia in humans by overcoming this regulation. Igf2bp3 is reported as an oncofetal gene that is expressed during embryonic stages and upregulated in cancers[44]. Our analysis also demonstrated that the expression of Igf2bp3 is downregulated during a neonatal stage in mice (Supplemental Fig. 6g). In the human leukemia specimens, KHDRBS3 is downregulated in pro-B cell leukemia carrying MLL-AF4 (Supplementary Fig. 6h). These lines of evidence suggest that the expression pattern of RBPs may be critical to circumvent the post-transcriptional regulation of MLL-AF4. The cell-of-origin of MLL-AF4 leukemia cells needs to have a unique combination of RBPs that enables the translation of MLL-AF4, which should be revealed in future.

In summary, we demonstrated that MLL-AF4 is post-transcriptionally regulated and that this post-transcriptional regulation critically influences its leukemogenic potential. Human *AF4*, but not murine *Af4*,

contains the motifs recognized by unique RBPs, including KHDRBSs. The same motifs are implicated in ribosomal stalling. An MLL-AF4 mutant with synonymous mutations transformed murine HSPCs and induced myeloid leukemia by evading the post-transcriptional regulation. Our findings indicate that the oncogenic activity of MLL-AF4 can be post-transcriptionally controlled by RBPs such as IGF2BP3 and KHDRBS families and suggest that modulating the functions of RBPs may lead to the development of novel therapies for this devastating disease.

## Methods

Our research complies with all relevant ethical regulations. Animal handling was approved by the National Cancer Center Institutional Animal Care and Use Committee and the Institutional Review Board of Yokohama City University (protocol no. F-A-20-043).

### Cells and cell culture

The human leukemia cell lines MV4-11 (ATCC) and SEM (DSMZ) were cultured in Iscove's modified Dulbecco's medium (IMDM) supplemented with fetal bovine serum (FBS; 10%) and penicillin-streptomycin (PS). The human leukemia cell lines, including RS4;11 (ATCC), HB1119[45], THP-1 (ATCC), CCRF-CEM (JCRB), CCRF-SB (JCRB), MOLT-4 (JCRB), KOPN-8 (DSMZ), KG-1 (DSMZ), EOL-1 (obtained from Michael Cleary), ML-2 (DSMZ), NOMO-1 (JCRB), and MOLM-13 (DSMZ) were cultured in Roswell Park Memorial Institute (RPMI) 1640 medium supplemented with 10% FBS and PS. The 293 T (ATCC) and 293TN (System Biosciences) cell lines were cultured in Dulbecco's modified Eagle's medium (DMEM) supplemented with 10% FBS and PS. Platinum-E retroviral packaging cell line (PLAT-E)[46] was cultured in DMEM supplemented with FBS (10%), puromycin (1 μg/mL), blasticidin (10 μg/mL), and PS. The murine stromal cell line MS-5 (DSMZ) was cultured in alpha-MEM supplemented with L-glutamine (2 mM), sodium pyruvate (2 mM), FBS (10%), and PS. Ba/F3 (RIKEN BRC) cells were cultured in RPMI 1640 medium supplemented with FBS (10%), PS, and murine interleukin-3 (mIL3; 10 ng/mL). Cells were cultured in the incubator at 37 °C and 5% CO2 and periodically tested for mycoplasma contamination using the MycoAlert Mycoplasma detection kit (Lonza). The details of the materials are described in Supplementary Table 1.

### Vector construction

The pMSCV-neo MLL ΔFP, MLL-ENL, and MLL-AF10 vectors have been described previously[7,22,47]. The fragments of human *AF4* and mouse *Af4* genes were generated using polymerase chain reaction (PCR) and cloned into the NruI and XhoI sites in the pMSCV-neo MLL fusion vectors. *AF4-MLL* gene was generated using PCR from the cDNA of MV4-11 and cloned into the BglII and EcoRI sites in the pMSCV-hygro vector using Gibson assembly master mix (NEB). The cDNAs were obtained via PCR using KOD-plus v2 DNA polymerase (TOYOBO) from 293 T cells, MV4-11 or murine embryonic fibroblasts[7]. Various gene constructs were generated using restriction enzyme digestion or PCR-based mutagenesis and cloned into the pMSCV-neo (for retrovirus production) (Clontech), pCDH-MSCV-MCS-EF1-Hygro (for lentivirus production/transient expression), or the pCMV5 vectors (for transient expression). The pCDH-MSCV-MCS-EF1-Hygro vector was generated via replacement of the puromycin-resistant gene by the hygromycin-resistant gene in the pCDH-MSCV-MCS-EF1-Puro vector (System Biosciences). sgRNA guide sequences were designed using CHOPCHOP[48] and were cloned into the BbsI-digested expression vectors, including the pX330-U6-Chimeric_BB-CBh-hSpCas9, pX335-U6-Chimeric_BB-CBh-hSpCas9n(D10A), and pKLV2-U6gRNA5(BbsI)-PGKpuro2ABFP-W vectors[49,50]. The shRNA sequences were designed using the GPP web portal (Broad Institute). The DNA fragments encoding the shRNA sequences were cloned into the AgeI- and EcoRI-digested pLKO.1-puro vector. The DNA fragments of FLAG-tagged GFP and HA-tagged RFP reporter genes were generated using PCR. The P2A sequence was sandwiched by HA-tagged RFP and FLAG-tagged GFP reporter genes and cloned into the XbaI and NotI sites

in the pCDH-MSCV-MCS-EF1-Hygro vector. For co-localization analysis with RBPs, the FLAG-tagged GFP reporter genes were cloned into EcoRI and NotI sites in the pCDH-MSCV-MCS-EF1-Hygro vector. The mCherry-tagged RBP expression vectors were generated using PCR and cloned into the pCDH-MSCV-MCS-EF1-Puro vector. For rescue experiments, the human IGF2BP3 gene was cloned in the pCDH-MSCV-MCS-EF1-copGFP vector (System Biosciences). The details of the materials are described in Supplementary Table 1. The sequences of the Oligonucleotides for gene editing and shRNA-mediated knockdown are described in Supplementary Table 2.

### Western blotting

Proteins were separated via sodium dodecyl sulfate-polyacrylamide gel electrophoresis (SDS-PAGE), followed by transblotting onto a polyvinylidene difluoride (PVDF) membrane using an iBlot 2 dry blotting system (Invitrogen). The membranes were blocked with iBind solution kit (Invitrogen) and incubated with primary antibodies suspended in the iBind solution (1 μg/ml or 1/1000 dilution, detailed concentration is available in Source Data file and Reporting Summary) overnight at 4 °C. The membranes were then washed thrice with phosphate buffered saline with 0.5% Tween-20 (PBS-T) and incubated with peroxidase-conjugated secondary antibodies for 1 h in the iBind solution (1/10000 dilution) at 25 °C. The blots were developed using an enhanced chemiluminescence reagent (GE Healthcare). For the western blotting of MLL fusion proteins, the proteins were separated using SDS-PAGE, followed by transblotting onto nitrocellulose membranes using a Mini-Protean Tetra cell (Bio-Rad Laboratories). The membranes were blocked with the iBind solution and incubated with primary antibodies suspended in the iBind solution overnight at 4 °C. The membranes were then washed thrice with PBS-T and incubated with peroxidase-conjugated secondary antibodies for 3 h in the iBind solution at 25 °C. The blots were developed using the enhanced chemiluminescence reagent. The details of the antibodies are described in Supplementary Table 1. Uncropped scans of western blotting data are available in Supplementary Information and Source Data file.

### Virus production

The ecotropic retrovirus was produced in PLAT-E packaging cells[46]. The lentivirus was produced in 293TN cells using the pMDLg/pRRE, pRSV-rev, and pMD2.G vectors[51]. The plasmids carrying transgenes were transfected using Lipofectamine 2000 reagent (Life Technologies) or polyethylenimine and the supernatant was replaced by fresh medium after 6 h of transfection. The supernatant containing the virus was harvested 24 to 48 h following transfection, filtered using a 0.45 μm PVDF membrane, and used for viral transduction. The details of the materials are described in Supplementary Table 1.

### Virus titration

The supernatant containing the virus particle in the packaging cell culture was harvested 48 h following transfection. The supernatant was filtered using a 0.45 μm PVDF membrane. The RNA derived from the viral genome was isolated using the RNeasy mini kit (Qiagen) and treated with in-column DNase digestion using the RNase-free DNase kit (Qiagen). The cDNA was synthesized using SuperScript III first-strand synthesis system with oligo (dT) primers (Invitrogen). The retrovirus concentration was determined using real-time qPCR with a TaqMan probe designed for the EPS region of the pMSCV vectors described in the Supplementary Table 1 (Life Technologies) and the absolute quantification method using the retrovirus plasmids DNA as references according to the ABI "absolute quantitation using standard curve getting started guide." The details of the materials are described in Supplementary Table 1.

### Assessment of virus genome integration

Bone marrow cells were harvested from the femurs and tibiae of 5-week-old female C57BL/6 mice. c-Kit-positive HSPCs were enriched

using magnetic beads conjugated with an anti-CD117 antibody (Miltenyi Biotech) and cultured in IMDM supplemented with FBS (10%), PS, murine stem cell factors (mSCF), mIL3, and interleukin-6 (mIL6) (10 ng/mL of each) for 24 h. In total, $5 \times 10^5$ HSPCs were then mixed with 5 mL of the supernatant containing recombinant retroviruses via spinoculation (1220 × $g$ at 32 °C for 2.5 h). After retrovirus transduction, the medium was replaced by IMDM supplemented with FBS (10%), PS, mSCF, mIL3, and granulocyte-macrophage colony-stimulating factors (mGM-CSF) (10 ng/mL) for 24 h. The genomic DNAs of the transduced cells were isolated using High Pure PCR template preparation kit (Roche). To assess virus genome integration in the Ba/F3 cells, this cell line was maintained in RPMI 1640 medium supplemented with FBS (10%), PS and mIL3 (10 ng/mL). The Ba/F3 ($1 \times 10^6$) cells were then incubated with 1 mL of the supernatant containing recombinant retrovirus for 24 h. After retrovirus transduction, the medium was replaced by RPMI 1640 medium supplemented with FBS (10%), PS, and mIL3 (10 ng/mL) and cultured for 24 h. The genomic DNAs of the Ba/F3 cells were isolated using the same method as HSPCs. The copy number of the virus genome was determined using real-time qPCR with MSCV-EPS TaqMan probes described in the Supplemental Table 3. The copy number of the virus genome normalized to that of the *Gapdh* locus was determined using a standard curve and the relative quantification method. The details of the materials are described in Supplementary Table 1.

## Myeloid progenitor transformation assay

Myeloid progenitor transformation assays were performed as described previously[15,21]. Bone marrow cells were harvested from the femurs and tibiae of five-week-old female C57BL/6 J mice. HSPCs were enriched using magnetic beads conjugated with an anti-CD117 antibody, transduced with recombinant retroviruses via spinoculation (1220 × $g$ at 32 °C for 2.5 h), and then plated in a methylcellulose medium (IMDM supplemented with 20% FBS, 1.6% methylcellulose, 100 μM β-mercaptoethanol, 10 ng/mL mSCF, 10 ng/mL mIL3, 10 ng/mL mGM-CSF, and PS). G418 (1 mg/mL) was added to the culture one day after the transduction to select for the transduced cells. For co-transduction with knockout constructs, HSPCs were transduced with a retrovirus carrying MLL-fusion and lentiviruses carrying Cas9 and sgRNAs using the RetroNectin-bound virus transduction method. RetroNectin-coated tubes were prepared by incubating with 100 μg/mL of RetroNectin (Takara Bio) overnight at 4 °C. The retrovirus and the lentiviruses were bound to the RetroNectin-coated tubes via centrifugation (1220 × $g$ at 32 °C for 2 h). The tubes were then washed with PBS, and the HSPCs were cultured overnight in the virus-bound tubes in IMDM supplemented with FBS (10%), mSCF (10 ng/mL), mIL3 (10 ng/mL), mGM-CSF (10 ng/mL), and PS in a CO$_2$ incubator at 37 °C. The next day, methylcellulose medium was added to the cells and plated. Antibiotics [G418 (1 mg/mL), blasticidin (10 μg/mL), and/or puromycin (1 μg/mL)] were added to the first round of culture to select for the transduced cells. For co-transduction of MLL-AF4 and AF4-MLL constructs, murine HSPCs were transduced with retroviruses carrying MLL-AF4 and AF4-MLL using the RetroNectin-bound virus transduction method as is the case with co-transduction with knockout constructs. Antibiotics [G418 (1 mg/mL), and hygromycin (800 μg/mL)] were added to the first round of culture to select for the transduced cells. *Hoxa9* expression was assessed using RT-qPCR after the first round of culture. Colony-forming units (CFUs) at the third and fourth rounds were determined per $10^4$ plated cells, after 5 days of culture. To evaluate the efficiency of the whole virus transduction cycle, the number of HSPCs was counted after 5 days of G418 selection and the whole cell lysate was prepared for western blot analysis. The details of the materials are described in Supplementary Table 1.

## Lymphoid progenitor transformation assay

To enable co-culture in the presence of G418, MS5-neo cells were established via transduction of MS5 cells with retrovirus carrying the pMSCV-neo vector. MS5-neo cells were maintained in alpha-MEM media supplemented with pyruvate, FBS (10%), PS, and G418 (1 mg/mL). HSPCs were harvested as described in the myeloid progenitor transformation assay. The cells were primed overnight using the cytokine treatment with mIL3, mSCF, and mIL6 and transduced with a recombinant retrovirus using spinoculation. The cells were then co-cultured with MS5-neo cells in IMDM supplemented with FBS (20%), mFlt3L (10 ng/mL), interleukin-7 (10 ng/mL), and PS. Cell number was counted after every 6 days. The details of the materials are described in Supplementary Table 1.

## In vivo leukemogenesis assay

In vivo leukemogenesis assays were performed as described previously[15,24]. HSPCs ($2 \times 10^5$) prepared from mouse femurs and tibiae were transduced with retrovirus using RetroNectin-bound virus transduction method as described in the myeloid progenitor transformation assay. Virus-infected cells were intravenously transplanted into sub-lethally irradiated (5–6 Gy) 7 weeks old female C57BL/6 J mice. Moribund mice were euthanized, and the cells isolated from bone marrow were analyzed using flow cytometry and freeze-stocked. For secondary transplantation, $2 \times 10^5$ freeze-stocked bone marrow cells were transplanted in the same manner. For the in vitro immortalized cell transplantation, $2 \times 10^5$ cells were transplanted in the same manner. This protocol was approved by the National Cancer Center Institutional Animal Care and Use Committee and the Institutional Review Board of Yokohama City University (protocol no. F-A-20-043). The details of the materials are described in Supplementary Table 1.

## Reverse transcription-quantitative PCR

RNA was prepared using the RNeasy kit and reverse-transcribed using a Superscript III First Strand cDNA synthesis kit with oligo(dT) primers. Gene expression was confirmed with qPCR, using the TaqMan probes described in Supplementary Table 1 (Life Technologies). The expression levels, normalized to those of *Gapdh*, were determined using a standard curve and the relative quantification method as described in the ABI User Bulletin #2.

## Identification of PTRS RNA-associated factors

The expression vector for FLAG-tagged RNA binding factor was transfected into 293 T cells using the Lipofectamine 2000 reagent. 293 T cells cultured in a 10 cm dish were trypsinized, washed with PBS, and collected. The collected cells were suspended in 1 mL of buffer A [150 mM NaCl, 10 mM Tris-HCl (pH 7.5), 1.5 mM MgCl$_2$, 0.5% NP-40, and an EDTA-free protease inhibitor cocktail (Roche)]. The suspension was centrifuged at 600 × $g$ for 3 min. The pellet was resuspended in 0.5 mL of buffer B [100 mM NaCl, 10 mM PIPES (pH 6.8), 3 mM MgCl$_2$, 1 mM EGTA, 0.3 M sucrose, 0.5% Triton X-100, 5 mM sodium butyrate, 0.5 mM dithiothreitol (DTT), 1 mM ribonucleoside vanadyl complex, and an EDTA-free protease inhibitor cocktail] and centrifuged at 600 × $g$ for 3 min. The supernatant was then transferred to a new tube (buffer B-fraction) and the pellet was resuspended in 0.5 mL of buffer C [50 mM Tris-HCl (pH 7.5), 4 mM MgCl$_2$, 1 mM CaCl$_2$, 0.3 M sucrose, 5 mM sodium butyrate, 0.5 mM DTT, and an EDTA-free protease inhibitor cocktail], followed by treatment with 1 unit micrococcal nuclease (MNase) for 7 min at 37 °C. The MNase reaction was stopped by adding EDTA (pH 8.0) to a final concentration of 20 mM. The buffer B-fraction was then added back to the buffer C fraction and cleared by centrifugation at 15,000 × $g$ for 5 min. The supernatant was transferred to a new tube, and 160 units of RNaseOUT recombinant ribonuclease inhibitor (Invitrogen) was added (nuclear fraction). Dynabeads M-280 streptavidin (10 μL; Invitrogen) equilibrated in 2X buffer C [10 mM Tris-HCl (pH 7.5), 1 mM EDTA, and 2 M NaCl] was added to biotin-conjugated RNA oligonucleotides (10 μL; 100 pmol; Invitrogen). The RNA oligonucleotide was fixed to the streptavidin beads for 15 min at 25 °C and washed with buffer C (1 mL). Next, 0.5 mL of the nuclear fraction was added to the

beads and incubated for 2 h with rotation at 4 °C. The beads were washed five times with 0.5 mL of buffer D [25 mM Tris-HCl (pH 7.5), 10 mM sodium phosphate, 125 mM NaCl, 2 mM MgCl$_2$, 0.5 mM CaCl$_2$, 0.15 M sucrose, 2.5 mM sodium butyrate, 0.75 mM DTT, 15 mM sodium pyrophosphate, 5 mM NaF, 10 mM EDTA, 0.05% NP-40, 5% glycerol, and an EDTA-free protease inhibitor cocktail]. The co-precipitated proteins were harvested in elution buffer (1% SDS, 50 mM NaHCO$_3$). The eluted samples were mixed with equal volume of 2X SDS-PAGE sample buffer and then subjected to silver staining and western blotting. For mass spectrometry analysis, the co-precipitated proteins were harvested in 1% RapiGest SF (Waters) at 67 °C for 5 min. The details of the materials are described in Supplementary Table 1.

### Competitive growth assay

Murine progenitor cells immortalized in myeloid progenitor transformation assay were cultured in IMDM supplemented with FBS (10%), PS, mSCF, mIL3, and mGM-CSF (10 ng/mL of each). The cells were transduced with a lentivirus co-expressing IGF2BP3 and GFP using the RetroNectin-bound virus transduction method as described in the myeloid progenitor transformation assay. The percentage of GFP-positive cells was analyzed using the FACSMelody cell sorter (BD Biosciences). The details of the materials are described in Supplementary Table 1.

### Reporter system for PTRS-dependent post-transcriptional regulation

The expression vectors for HA-tagged RFP/FLAG-tagged-GFP-PTRS reporters were transfected into 293 T cells using the Lipofectamine 2000 reagent. 293 T cells were then incubated in a CO$_2$ culture chamber for 24 h. The medium was then replaced by FluoroBrite DMEM (Thermo Fisher Scientific) with FBS (10%). Images were visualized using a BZ-X710 microscope (KEYENCE), and the average fluorescence intensity was analyzed using ImageJ software. For co-localization analysis of RBPs, expression vectors for the FLAG-tagged-GFP-hPTRS reporter and mCherry-tagged-RBPs were co-transfected into 293 T cells using Lipofectamine 2000 reagent. After 24 h, the medium was replaced by FluoroBrite DMEM supplemented with FBS (10%), followed by the addition of Hoechst 33342 dye (5 μg/mL; Thermo Fisher Scientific). Images were visualized using the BZ-X710 microscope. The details of the materials are described in Supplementary Table 1.

### Identification of PTRS reporter-bound proteins

Expression vectors carrying HA-tagged RFP/FLAG-tagged-GFP-PTRS reporters were transfected into 293 T cells using the Lipofectamine 2000 reagent. 293 T cells cultured in a 10 cm dish were trypsinized, washed with PBS, and collected. The collected cells were suspended in 1 mL of NIB buffer [10 mM Tris-HCl (pH 7.5), 5 mM MgCl$_2$, 0.32% sucrose, 1% Triton X-100, and an EDTA-free protease inhibitor cocktail]. The suspension was centrifuged at 2000 × $g$ for 3 min. The pellet was resuspended in 1 mL of RIP buffer [25 mM Tris-HCl (pH 7.4), 150 mM KCl, 0.5 mM DTT, 5 mM EDTA, 0.5% NP-40, and an EDTA-free protease inhibitor cocktail] and centrifuged at 15,000 × $g$ for 5 min. The supernatants were transferred to new tubes and subjected to immunoprecipitation using FLAG-M2 magnetic beads (Sigma-Aldrich). The beads were washed five times with 0.5 mL of RIP buffer and the co-precipitated proteins were harvested in elution buffer. The eluted samples were mixed with equal volume of 2X SDS-PAGE sample buffer and then subjected to western blotting. For mass spectrometry analysis, 293 T cells expressing FLAG-tagged-GFP-PTRS reporters were suspended in 1 mL of buffer B and subjected to immunoprecipitation using FLAG-M2 magnetic beads. The beads were washed five times with 0.5 mL of buffer B and the coprecipitated proteins were harvested in 1% RapiGest SF. The details of the materials are described in Supplementary Table 1.

### Two-step IP analysis

The expression vector for FLAG-tagged RNA binding factor was transfected into 293 T cells using the Lipofectamine 2000 reagent. The cell lysate was prepared by the same methods described in the section of Identification of PTRS RNA-associated factors. The biotin-conjugated RNA oligonucleotides were added to the lysate, and then subjected to immunoprecipitation using FLAG-M2 magnetic beads for 2 h with rotation at 4 °C. The beads were washed five times with 0.5 mL of buffer D. The precipitated proteins were harvested in FLAG elution buffer [10 mM Tris-HCl (pH 8.0), 150 mM NaCl, 1 mM EDTA, 5 mg/ml 3xFLAG peptide, and an EDTA-free protease inhibitor cocktail] by incubating for 5 mins at 4 °C. Next, streptavidin beads were added to the eluted fraction and incubated for 1 h with rotation at 4 °C. The streptavidin beads were washed five times with 0.5 mL of buffer D. The co-precipitated proteins were harvested in elution buffer (1% SDS, 50 mM NaHCO$_3$). The eluted samples were mixed with equal volume of 2X SDS-PAGE sample buffer and subjected to western blotting. 1% equivalent amount of the cell lysate after the addition of biotin-conjugated RNA oligonucleotides was prepared as input samples.

### Actinomycin D and cycloheximide chase analysis

Expression vectors carrying HA-tagged RFP/FLAG-tagged-GFP-PTRS reporters were transfected into 293 T cells using the Lipofectamine 2000 reagent. For actinomycin D chase analysis, actinomycin D (10 μg/mL) was added to the culture medium of the 293 T cells and the RNAs were prepared at the indicated time points. RNA was prepared using the NucleoSpin RNA kit and reverse-transcribed using a PrimeScript RT reagent kit. Gene expression was confirmed with qPCR, using the SYBR green methods with the primers described in Supplementary Table 4. The expression levels, normalized to those of *GAPDH*, were determined using a standard curve and the relative quantification method. For cycloheximide chase analysis, cycloheximide (100 μg/mL) was added to the culture medium of the 293 T cells and the total cell lysates were prepared at the indicated time points. The reporter proteins were analyzed by western blotting.

### Flow cytometry

Bone marrow cells were harvested from the femurs and tibiae of leukemic mice. Red blood cells were removed via treatment with ACK lysis buffer (150 mM NH$_4$Cl, 10 mM KHCO$_3$, and 0.1 mM EDTA), and then cells were stained with antibodies in PBS with 3% FBS (1/100-1/500 dilution, detailed concentration is available in Reporting Summary), and analyzed using the FACSMelody cell sorter or FACSCelesta flow cytometer. The antibodies are described in Supplementary Table 1.

### LC-MS /MS

Trypsinization of proteins was performed as previously described[52,53]. Tandem MS (MS/MS) analysis was performed using an LTQ Orbitrap ELITE ETD mass spectrometer (Thermo Fisher Scientific). The mass spectrometer was operated in data-dependent acquisition mode, in which MS acquisition with a mass range of 400 to 1000 m/z was automatically switched to MS/MS acquisition under the control of the Xcalibur software (Thermo Fisher Scientific). The top four precursor ions in the MS scan were selected using Orbitrap, with a resolution of R = 240,000, and the ions in subsequent MS/MS scans were analyzed with an ion trap in automated gain control (AGC) mode, in which AGC values were $1 \times 10^6$ and $1.00 \times 10^4$ for full MS and MS/MS, respectively. Electron transfer dissociation (ETD) was used for fragmentation.

### mRNA-sequencing

Total RNA from freeze stocked bone marrow cells of leukemic mice was prepared using the RNeasy kit and the quality was assessed using a

RNA6000 Pico Kit (Agilent) in 2100 Bioanalyzer (Agilent). The cDNA was synthesized and amplified with the Sure Select Strand Specific RNA Library Prep Kit (Agilent Technologies). Paired-end 37 bp sequencing was conducted on the NextSeq 500 platform (Illumina). Sequence reads were aligned to mouse reference genome (GRCm39) in STAR software. The expression was quantified using RSEM software. The count data was normalized using TMM method in the edgeR package and heat maps were constructed in the heatmap package under R environment.

## Public single cell RNA-seq analysis

The blood single cell RNA-seq data of murine fatal HSPCs was downloaded from Gene Expression Omnibus (GSE128761) and analyzed in the Seurat package under R environment. The microarray data of human leukemia specimens was downloaded from Gene Expression Omnibus (GSE13164) and the log2 scaled probe intensities were plotted using Prism 9.3 software (GraphPad Software Inc.).

## Statistics and reproducibility

Statistical analysis was performed using GraphPad Prism 9.3 software. Data are presented as the mean ± standard deviation (SD) of at least three biologically independent experiments. Two groups were compared using the unpaired two-tailed Student's $t$-test, while multiple comparisons were performed using analysis of variance (ANOVA) followed by Tukey's test. The details of the software are described in Supplementary Table 1. All the experiments are independently performed at least twice and confirmed their reproducibility.

## Reporting summary

Further information on research design is available in the Nature Portfolio Reporting Summary linked to this article.

## Data availability

All data supporting the findings of this study are available within the article, in the supplementary information, and in the source data. The RNA-seq data generated in this study have been deposited in Gene Expression Omnibus under accession code GSE201503. The raw data of mass spectrometry have been deposited in the JPOST repository (https://repository.jpostdb.org/entry/JPST001132). Further information and requests for resources and reagents should be directed to and will be fulfilled by Hiroshi Okuda (okuda.hir.tv@yokohama-cu.ac.jp) or Akihiko Yokoyama (ayokoyam@ncc-tmc.jp). Source data are provided with this paper as a Source Data file.

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

## Acknowledgements

We thank Makiko Okuda, Yuzo Sato, and Ayako Yokoyama for their technical assistance. We thank Dr. Toshio Kitamura for providing the PLAT-E cells and Dr. Michael L. Cleary for providing the HB1119 and EOL1 cells. We also thank Shonai Regional Industry Promotion Center members for their administrative support. This work was supported by JSPS KAKENNHI (grants 17H07379 and 20K08722 to H.O.; 16H05337 and 19H03694 to A.Y.) and the MHLW program (grant H23-Seisakutansaku-wakate-001 to A.Y.). This research was also supported by MEXT Promotion of Distinctive Joint Research Center Program Grant Numbers JPMXP0618217493, JPMXP0622717006 at the Advanced Medical Research Center, Yokohama City University. This work was also supported in part by research funds from the Yamagata prefectural government, the City of Tsuruoka, and Sumitomo Pharma.

## Author contributions

Conceptualization: H.O., A.Y.; Methodology: H.O., R.M., S.T., J.I., I.H., AY.; Investigation: H.O., R.M., S.T., J.I., I.H., A.Y.; Data Curation: K.T.; Writing–original draft: H.O., A.Y.; Writing–review & editing: H.O., A.Y.; Funding acquisition: H.O., A.Y.; Supervision: H.O., T.M., A.Y.

## Competing interests

A.Y. received a research grant from Sumitomo Pharma. The other authors have no competing interests.
