## [Peer Review File · Nature Communications]

Reviewers' Comments:

Reviewer #1:

Remarks to the Author:

In this paper, Okada et al set out to determine why MLL-AF4 is not able to transform mouse cells. They found that rather than viral production being the major inhibitor of MLL-AF4 transforming capabilities, a short sequence in human AF4 was responsible for translation of the viral RNA. They found that an MLL-AF4 mutant PTRS could transform mouse cells and produce an AML, with a few rare mixed lineage leukemias. They then went on to purify proteins that bound to the human (but not mouse) sequence, and identified several RBPs, including IGF2BP3. IGF2BP3 knockouts combined with deletions of KHDRBS binding sites (ie. AU1/2) allowed MLL-AF4 to transform mouse cells. Finally, they showed that the PTRS sequence likely functions through ribosomal stalling.

This is a very interesting paper with very clear and well-done experiments. I have a few questions and comments.

1. It might be important to show that the MLL-AF4 sAU13/Igf2bp3 colony assays (Fig. 5a) can also produce a leukemia in mice. Sometimes cells can be transformed, but are unable to engraft in a mouse.
2. Western blots for Igf2bp3 should be performed in the rescue experiment in Figure 5c.
3. This same post transcriptional regulation of MLL-AF4 seems to function in human cells (the 293 data), so why is MLL-AF4 not degraded in patients? One might think that this degradation/turnover mechanism might prevent leukemia in humans. The authors propose that it could be due to a specific cellular context being selected for. Does overexpression of IGF2BP3 kill human MLL-AF4 cell lines? Does it have any effect on other MLL fusion protein lines?

Reviewer #2:

Remarks to the Author:

Review of manuscript #NCOMMS-21-11746, entitled "RNA-Binding Proteins Control the Oncogenic Activity of MLL-AF4"

by Okuda et al.,

The authors describe in their manuscript a profound body of work dealing with a problem that hasn't been solved for nearly 2 decades. The most frequent t(4;11) translocation, associated with proB ALL in infants, pediatric and adult patients, is causing the production of two reciprocal fusion proteins, namely MLL-AF4 and AF4-MLL. Based on comparable translocations (e.g. MLL-ENL or MLL-AF9), many scientists were working in the past only with retro- or lentiviral MLL-AF4 constructs to model the disease on the murine system. But they all failed to obtain leukemia in mice. Very recently, a group in the USA could model for the first time the human disease phenotype in murine (AML) and human hematopoietic stem cells (preB ALL) by exchanging the human AF4 C-terminal portion with the homologous mouse Af4 sequence. This puzzled many scientists, but Okuda et al. is now bringing up a potential explanation for all these experimental results.

Noteworthy, work with AF4-MLL - even without MLL-AF4 also produced a B/T ALL, and recent experiments using the CRISP/R system to create balanced chromosomal translocations in human cord blood hematopoietic stem cells, producing both fusion proteins, were also successful. Nevertheless, Okuda et al. have solved in the present manuscript a long-lasting problem in such an elegant way, that I can only congratulate them for their success.

Basically, the authors have found three small motifs in the mRNA of the C-terminal coding portion

of AF4 that are only in the human sequence - not in the murine counterpart - binding specific RNA binding proteins that lead to a stalling of the translation machinery. To this end, although mRNA is produced, the amount of protein is so low that no transformation can occur in their experimental system. When this sequence motif is replaced by the murine counterpart, or mutated, they found in their assays transformation capacity and in their in vivo work the development of AML in mice.

Criticism/Comments

1. The authors do not explain, why this translocation in human cells - although the translational inhibitor sequence is present in the MLL-AF4 mRNA is able to cause leukemia. Following the arguments of the authors, this should not happen, because of the inhibitory sequence. Can the authors explain this by their experiments? What about the expression of IGF2BP3 in the human hematopoietic system? Or fetal liver cells? Maybe the authors can discuss this in their paper as well.
2. Others have shown that AF4-MLL was also able to cause B/T leukemia in murine Lin-negative Flt3/Sca1+ hematopoietic stem cells. How does this fit into their model about MLL-AF4 as sole oncoprotein?
3. The authors are aware of recent publication where CRISPR/Cas9-mediated t(4;11) translocation in human cord blood cells is also causing a leukemia phenotype. Also here, translation of the MLL-AF4 should be compromised and no transformation should occur if the authors are right. But this is not the case. If both fusion proteins are present, no such phenomenon occurs. To this end, is AF4-MLL able to overcome the inhibitory mechanism to produce MLL-AF4 protein?
4. In my opinion, the authors have found an important mechanism that explain the translational behavior of the human MLL-AF4 expression construct in murine and human cells. However, can they also explain the results of the Caslini paper of 2004 in Leukemia, where increased expression of MLL-AF4 caused a cell cycle arrest instead of transformation?
5. And this leads me to the final question: if translation of the mLL-AF4 mRNA is impaired, can this be out-titrated by increasing simply the mRNA in cells?

Overall this is a technical and an intellectual masterpiece of work.

Reviewer #3:

Remarks to the Author:

In the paper by Hiroshi Okuda et al, the authors explore mechanisms of MLL-AF4 and MLL-mAf4-mediated leukemogenesis using immune-competent mouse models of leukemia. Understanding mechanisms of MLL-AF4 leukemogenesis is an important area of research. Successful modeling of MLL-AF4 B-ALL will be helpful for the development of novel and improvement of existing treatments.

The major claim of this paper is that oncogenic activity of MLL-AF4 is controlled by RNA-binding proteins (RBPs) post-transcriptionally in such a way that RBPs repress MLL-AF4 protein production (translation). The authors could rescue the expression of MLL-AF4 and induce AML but not B-ALL in mouse HSCs. The author's claim that they found a minimal 54pb fragment within human AF4 sequence responsible for translational silencing of MLL-AF4. This post-transcriptional regulatory sequence (PTRS) of human AF4 contains binding sites for IGF2BP and KHDRBS family of proteins. Synonymous mutations disrupting the association with KHDRBS resulted in proper translation MLL-AF4 and induced AML in vivo. These findings are novel and potentially are of interest to others in the scientific community, however major concerns about correctness of some claims exist and must be addressed.

Main points

1. In this study, Okuda et al. had a working hypothesis that "RBPs have a potential to specifically

inactivate this (MLL-AF4) oncogene". The major concern is that authors did not take an unbiased approach to investigate trans-acting regulatory elements of MLL-AF4 transcript in mouse and human cells. Although their work of identification of cis-acting RNA elements with a number of mutants is properly done and results are convincing, identification of trans-acting elements regulating activity of MLL-AF4 was restricted to proteins and not RNA elements. For example, the role of non-coding RNA (e.g., miRNA, lncRNA) was not investigated and discussed. The authors rightly noted in Fig 1h that MLL-AF4 RNA levels are lower than other constructs which may indicate rapid mRNA decay. However the mRNA and protein stability before and after PTRS modifications was not assessed (actinomycin D and cycloheximide treatments respectively). The PTRS modifications may affect miRNA binding sites, stabilize RNA, and increase MLL-Af4 protein levels. The recommendation would be to screen the minPTRS for probable miRNAs and apply either anti-miR or mimics to attenuate mRNA stability and translation. It has been shown that IGF2BP/CRD-BP binds mRNA coding regions and protects coding region from miRNA-mediated degradation.

2. The most confusing and controversial part of the study is the role of Igf2bp3 as an inhibitor of MLL-AF4 translation and leukemia initiation. IGF2BPs are known as oncogenes promoting mRNA stability and translation. IGF2BPs were shown to be overexpressed in MLL rearranged human leukemia and, most likely, Igf2bp3 knockout in MLL-mAf4 leukemia will decrease its aggressiveness. Therefore, the claim "IGF2BP3 is responsible for the post-transcriptional inactivation of MLL-AF4" requires the most careful investigation. Besides in vitro, functional in vivo tests for the role of Igf2bp3 in MLL-Af4, MLL-AF4 (MLL-ENL, MLL-AF9 as controls) should be done. What effect do authors see with Igf2bp3 KO in MLL-mAf4 and other MLLr leukemia? Is it critical to show the role of Igf2bp3 in other types of MLLr leukemia and IGF2BP3 in human samples with MLL-AF4. The additional/alternative to what authors use in this paper gene knockout/ knockdown techniques should be used for IGF2BP3 gene silencing. It is not unusual for post-transcriptional regulators to have a dual role of oncogenes or tumor suppressors in different species. The non-coding RNAs are the most important player between different species like human and mouse as protein-coding sequences are highly homologous. Thus, the authors should investigate (and provide a solid evidence) if this is a mouse-specific mechanism where Igf2bp3 suppresses expression of human MLL-AF4, mMLL-hAF4, but not hMLL-mAf4, and indicate the role of these cis-acting elements in human leukemia.

3. The conclusion about MLL-AF4 translation inhibited by ribosomal stalling which occurs at AU-rich elements recognized by KHDRBS (in the abstract and discussion) is confusing as the authors show that IGF2BP3 is responsible for "the post-transcriptional inactivation of MLL-AF4". While IGF2BPs are cytoplasmic, KHDRBS are located mostly in the nucleus. Functional screen for RBPs identifying Igf2bp3 as an inhibitory factor of AF4 also raises a question about the role of other indicated proteins in regulation of MLL-AF4 and their interplay, co-localization etc. The conclusion "MLL-AF4 is post-transcriptionally regulated, which explains why the establishment of a mouse disease model by MLL-AF4 has been difficult" is very general because literally all transcripts are bound to RBPs and are post-transcriptionally regulated. The manuscript would benefit from focusing on one protein with a proper characterization of RNA-protein interaction including the full-length mRNA transcripts.

4. Identification of the cis-acting element responsible for induction of leukemia, even though it is not B-ALL, is a progress. However, the authors should clearly indicate what kind of leukemia was induced and what phenotypes they couldn't induce throughout the paper including abstract. RNA seq or ChIPseq would be a great addition. A cell type – mouse or human – used in manipulations should be mentioned more often through their manuscript including abstract. Information about spontaneous leukemia is irrelevant and should be removed from the manuscript.

Additional Notes:

1. RNA-binding proteins specifically associate with the PTRS of human AF4: you investigate protein binding with 210 or minimal 54bp fragment without a UV crosslink?
2. Many RBPs (e.g., KHDRBS1/3, ELAVL1) specifically bound to hPTRSmin – in fact, many RBPs recognize similar or same primary and secondary RNA structures and compete for binding sites. Not only IGF2BPs can attenuate translation, stability etc.. The MLL-AF4 regulatory element you have identified is embedded in RNA.

Point-by-point response to REVIEWER COMMENTS

Reviewer #1 (Remarks to the Author):

1. It might be important to show that the MLL-AF4 sAU13/Igf2bp3 colony assays (Fig. 5a) can also produce a leukemia in mice. Sometimes cells can be transformed, but are unable to engraft in a mouse.

Response: To answer to the reviewer #1's question, we transplanted MLL-AF4 sAU13/Igf2bp3-immortalized cells into syngeneic recipient mice. As shown in Supplementary Fig.5e, two of the recipient mice died during the one year observation period. These two mice were found dead in a condition where postmortem analysis was not possible, and thus, the cause of death could not be determined. In this experimental setting, we observed spontaneous B/T-ALL leukemia cases in a long latency (3 of 10 in this set of experiment), which were not caused by the transplanted immortalized cells because the transgene was not detected by RT-PCR. Thus, we concluded that MLL-AF4 sAU13/Igf2bp3-immortalized cells did not develop leukemia during the one-year period (Supplementary Fig.5e).

Generally, weak oncogenes are inefficient in developing leukemia *in vivo*; for example, MLL-ELL, for which the average colony forming units (CFU) is around 100 colonies/10,000 cells, induced leukemia inefficiently (with a median latency of 6 months and 20% penetrance), while MLL-AF10, for which the average CFU is approximately 1,500 colonies/10,000cells, induced leukemia within three months at 100% penetrance in our previous study¹. The average CFU of MLL-AF4 sAU13/Igf2bp3-immortalized cells is less than 100 colonies/10,000 cells (Fig. 5a), which falls into the class of least proliferative immortalized cells. Thus, we think that MLL-AF4 sAU13/Igf2bp3-immortalized cells did not have adequate proliferative capacities to induce leukemia *in vivo* in this experimental condition. At this point, it is technically difficult to examine whether these immortalized cells can also produce a leukemia in mice *in vivo*.

2. Western blots for Igf2bp3 should be performed in the rescue experiment in Figure 5c.

Response: We performed a western blotting experiment in 293T cells stably expressing the FLAG-tagged IGF2BP3 using the lentiviral gene transfer method (Figure R1), confirming that transgenes can be expressed through this method. However, the exogenously expressed IGF2BP3 protein could not be detected in the MLL-AF4 sAU13-immortalized Igf2bp3-knockout murine cells, likely because the cells expressing IGF2BP3 showed tendency to die and therefore, were not applicable for the detection by western blotting.

Figure R1 Western blotting of FLAG-tagged IGF2BP3 in 293T cells. FLAG-tagged IGF2BP3 was detected by anti-FLAG antibody.

3. This same post transcriptional regulation of MLL-AF4 seems to function in human cells (the 293 data), so why is MLL-AF4 not degraded in patients? One might think that this degradation/turnover mechanism might prevent leukemia in humans. The authors propose that it could be due to a specific cellular context being selected for. Does overexpression of IGF2BP3 kill human MLL-AF4 cell lines? Does it have any effect on other MLL fusion protein lines?

Response: To answer the reviewer #1's question, we examined the effects of overexpression of various RBPs in human leukemia cell lines, MV4-11 (MLL-AF4 cell line) and THP1 (MLL-AF9 cell line) (Figure R2)

Figure R2 Effects of over-expression of RNA binding proteins on cell growth of human leukemia cell lines carrying MLL-fusion genes. Data are shown as the mean \pm SD of three biological replicates.

Overexpression of single RBPs including IGF2BP3 did not kill the leukemia cells or arrest of the cell cycle, suggesting that simply increasing the expression level of RBP in a cell that has overcome this inhibitory

mechanism is not sufficient to trigger the post-transcriptional regulatory mechanism. We speculate that combinatorial functions of multiple RBPs are necessary to alter the regulatory mechanism. This view is supported by the observations that simply knocking out *Igf2bp3* does not confer transforming capacity to MLL-AF4 (Supplementary Fig.5a, b).

Reviewer #2 (Remarks to the Author):

1. The authors do not explain, why this translocation in human cells - although the translational inhibitor sequence is present in the MLL-AF4 mRNA is able to cause leukemia. Following the arguments of the authors, this should not happen, because of the inhibitory sequence. Can the authors explain this by their experiments? What about the expression of IGF2BP3 in the human hematopoietic system? Or fetal liver cells? Maybe the authors can discuss this in their paper as well.

Response: How can MLL-AF4 cause leukemia despite of the inhibitory mechanism described here? That's the question that we hope to answer eventually. In this study, we answered the question why MLL-AF4 cannot cause leukemia in murine disease models whereas other MLL fusions can, which has been an enigma in the field for a long time. We believe this study provides a valuable piece of information toward understanding the molecular mechanism of MLL-AF4-mediated leukemia. Here, we demonstrated that MLL-AF4 is subject to a unique post-transcriptional regulatory mechanism, which may play an important role to answer why MLL-AF4 preferentially induces infant lymphoblastic leukemia. Although we tested several possibilities in the previous version and in this revision, we haven't found a clear answer as to why this translocation in human cells is able to cause leukemia. The cell-of-origin may be implicated in the context which MLL-AF4 can be stably translated. We have discussed this in the discussion section of the revised manuscript (pages 21–22).

In the human leukemia samples of Pro-B ALL with t(4;11) translocation (MLL-AF4), *IGF2BP2/3* were highly expressed, whereas *KHDRBS3* expression was downregulated (see Supplementary Fig. 6h). We analyzed *Igf2bp3* expression in murine embryonic to post-natal hematopoietic stem cells and hematopoietic progenitor cells using publicly available data. *Igf2bp3* expression was high in utero and gradually reduced after birth (see Supplementary Fig. 6g). The population of detectable *Igf2bp3*-expressing cells was below 40% at each time point, suggesting that significant portions of the hematopoietic cells did not express *Igf2bp3*. These results suggests that MLL-AF4 leukemia cells may arise selectively in an IGF2BP3-deficient context, but evolve to accustom to the IGF2BP3-expressing cellular context during the development of MLL-AF4-leukemia.

2. Others have shown that AF4-MLL was also able to cause B/T leukemia in murine Lin-negative Flt3/Sca1+ hematopoietic stem cells. How does this fit into their model about MLL-AF4 as sole oncoprotein?

Response: AF4-MLL genes that are fused in-frame are often generated as a result of chromosomal translocations. It has been shown that AF4-MLL caused leukemia in murine disease models², suggesting an

oncogenic role for AF4-MLL as reviewer #2 mentioned. Thus, we constructed expression vectors for the AF4-MLL gene and performed myeloid progenitor transformation assay (see Supplemental Fig. 2e). However, in our assay conditions, no colony was observed in cells transduced with AF4-MLL (see Supplemental Fig. 2e). These data suggest that MLL-AF4, but not AF4-MLL, is the major oncogenic driver that can be evaluated in our experimental system.

3. The authors are aware of recent publication where CRISPR/Cas9-mediated t(4;11) translocation in human cord blood cells is also causing a leukemia phenotype. Also here, translation of the MLL-AF4 should be compromised and no transformation should occur if the authors are right. But this is not the case. If both fusion proteins are present, no such phenomenon occurs. To this end, is AF4-MLL able to overcome the inhibitory mechanism to produce MLL-AF4 protein?

Response: We are aware of the studies showing that CRISPR/Cas9-mediated translocation in human cord blood cells caused leukemia^{3,4}. The studies indicate that the cell-of-origin in the human cord blood cells provides a cellular context where MLL-AF4 can be expressed. We have mentioned this in the discussion section. It is an interesting suggestion that the combinatorial expression of MLL-AF4 and AF4-MLL might cancel the inhibitory effect described here. Thus, we performed myeloid progenitor transformation assay using a combination of the MLL-AF4 and the AF4-MLL (see Supplemental Fig. 2g). However, no colonies were observed by the co-expression of AF4-MLL with MLL-AF4 in that experiment. Thus, at this point, we don't think that AF4-MLL plays critical roles in the regulatory mechanisms of MLL-AF4.

4. In my opinion, the authors have found an important mechanism that explain the translational behavior of the human MLL-AF4 expression construct in murine and human cells. However, can they also explain the results of the Caslini paper of 2004 in Leukemia, where increased expression of MLL-AF4 caused a cell cycle arrest instead of transformation?

Response: We think it is possible that the overexpression of MLL-AF4 induces cell cycle arrest in hematopoietic progenitors. We observed a very low cell number after transduction of MLL-AF4 into murine hematopoietic progenitors (Fig. 1g). It is an interesting aspect of MLL-AF4-mediated functions that needs to be addressed in future.

5. And this leads me to the final question: if translation of the MLL-AF4 mRNA is impaired, can this be out-titrated by increasing simply the mRNA in cells?

Response: To answer the reviewer #2's question, we constructed MLL-AF4 expressing plasmids driven by pCMV promoter (pCMV5 MLL-AF4) and transfected them into 293T cells. The MLL-AF4 protein was detected when its expression was driven by the CMV promoter, whereas it is not when driven by the LTR of MSCV (see Supplementary Fig. 1c). These results suggest that it is possible to titrate out the inhibitory RBPs by increasing the mRNA levels of *MLL-AF4*.

Reviewer #3 (Remarks to the Author):

Main points

1. In this study, Okuda et al. had a working hypothesis that “RBPs have a potential to specifically inactivate this (MLL-AF4) oncogene”. The major concern is that authors did not take an unbiased approach to investigate trans-acting regulatory elements of MLL-AF4 transcript in mouse and human cells. Although their work of identification of cis-acting RNA elements with a number of mutants is properly done and results are convincing, identification of trans-acting elements regulating activity of MLL-AF4 was restricted to proteins and not RNA elements. For example, the role of non-coding RNA (e.g., miRNA, lncRNA) was not investigated and discussed. The authors rightly noted in Fig 1h that MLL-AF4 RNA levels are lower than other constructs which may indicate rapid mRNA decay. However the mRNA and protein stability before and after PTRS modifications was not assessed (actinomycin D and cycloheximide treatments respectively). The PTRS modifications may affect miRNA binding sites, stabilize RNA, and increase MLL-Af4 protein levels. The recommendation would be to screen the minPTRS for probable miRNAs and apply either anti-miR or mimics to attenuate mRNA stability and translation. It has been shown that IGF2BP/CRD-BP binds mRNA coding regions and protects coding region from miRNA-mediated degradation.

Response: To assess the possibilities of the trans-acting regulatory elements of the *MLL-AF4* transcript, we performed a motif search for the recognition sequences of miRNAs and lncRNAs within the PTRSs using two different databases: miRbase and lncRNADB. Within the minimum hPTRS, we found no recognition motifs for miRNA and lncRNA (see Supplementary Figure 4c). To respond to the reviewer #3’s concern, we also assessed RNA and protein stability using actinomycin D- and cycloheximide-chase methods in our reporter system. The stabilities of RNAs were comparable among the wild type and mutants (see Supplementary Fig. 6b). In contrast, the stability of the PTRS-fused GFP protein was slightly lower compared to that of the wild type. However, the kinetics of the degradation of the PTRS-fused reporter proteins were comparable among the proteins (see Supplementary Fig. 6c). These data indicate that RNA-binding proteins, but not non-coding RNAs are the main contributors to the regulation of MLL-AF4 expression in this context.

2. The most confusing and controversial part of the study is the role of IGF2BP3 as an inhibitor of MLL-AF4 translation and leukemia initiation. IGF2BPs are known as oncogenes promoting mRNA stability and translation. IGF2BPs were shown to be overexpressed in MLL rearranged human leukemia and, most likely, IGF2BP3 knockout in MLL-mAf4 leukemia will decrease its aggressiveness. Therefore, the claim “IGF2BP3 is responsible for the post-transcriptional inactivation of MLL-AF4” requires the most careful investigation. Besides in vitro, functional in vivo tests for the role of IGF2BP3 in MLL-Af4, MLL-AF4 (MLL-ENL, MLL-AF9 as controls) should be done. What effect do authors see with IGF2BP3 KO in MLL-mAf4 and other MLLr leukemia? Is it critical to show the role of IGF2BP3 in other types of MLLr leukemia and IGF2BP3 in human samples with MLL-AF4. The additional/alternative to what authors use in this paper gene knockout/knockdown techniques should be used for IGF2BP3 gene silencing. It is not unusual for post-transcriptional regulators to have a dual role of oncogenes or tumor suppressors in different species. The non-coding RNAs

are the most important player between different species like human and mouse as protein-coding sequences are highly homologous. Thus, the authors should investigate (and provide a solid evidence) if this is a mouse-specific mechanism where *Igf2bp3* suppresses expression of human MLL-AF4, mMll-hAF4, but not hMLL-mAf4, and indicate the role of these cis-acting elements in human leukemia.

Response: During this revision, Tran et al. have reported that IGF2BP3 has pro-oncogenic roles in MLL-Af4-mediated leukemia⁵. To answer to the reviewer #3's concern, we first evaluated the roles of IGF2BP3 in MLL-fusion mediated leukemogenesis both ex vivo and in vivo. In the ex vivo myeloid transformation assay, knockout of *Igf2bp3* resulted in a mild decrease of colony forming ability of MLL-mAf4 (which is partly consistent with what was reported by Tran et al.), but not of mMll-mAf4 or MLL-AF10 (see Supplementary Fig. 5d). Next, we also evaluated the role of IGF2BP3 in vivo in MLL-AF10-mediated leukemia. We transplanted IGF2BP3-deficient MLL-AF10-immortalized cells into syngeneic mice, but did not observe any recognizable anti-tumorigenic effects caused by *Igf2bp3* knockout (see Supplementary Fig. 5e). Thus, our results do not fully support the notion of IGF2BP3 as a pro-oncogenic factor. However, we do not exclude the possibility that IGF2BP3 has some pro-oncogenic functions featured in MLL-AF4-mediated leukemia described by Tran et al. IGF2BP3 may function as a bi-directional factor for oncogenesis by suppressing MLL-AF4 translation but promoting oncogenesis by modulating its target gene transcription/splicing, which we have mentioned in the discussion section (page 23 line 5).

As for the implication of non-coding RNA, we did motif searches for non-coding RNA within the minPTRS and did not find any of those in hPTRS as described above. Thus, we focused on RBPs instead of non-coding RNAs in this study. The post-transcriptional regulation of AF4 was observed both in mouse (HPSCs) and human cells (293T) as shown in this study. Because simple knockout or overexpression of RBPs did not drastically alter the AF4 regulation, demonstrating the role of each RBPs has been challenging. As shown in this study, simple knockout of *Igf2bp3* did not confer stable expression of MLL-AF4. The mutations of AU-rich sites were additionally required. We think this will be a good starting point to address the reviewer #3's (and our) questions in the future, but it is out of scope to fully elucidate the mechanism in one manuscript.

To evaluate the potential roles of cis-acting elements on the *MLL* genes, we cloned the murine *Mll* gene and constructed plasmids carrying genes in combinations of murine *Mll*/human *MLL* and murine *Af4*/human *AF4*. In the results of the myeloid progenitor transformation assay using the constructs, vigorous colonies were observed in MLL-mAf4 and mMll-mAf4, whereas no colony was observed in MLL-AF4 and mMll-AF4, suggesting that posttranscriptional regulation takes place specifically on human AF4 (see Supplementary Fig. 5d).

3. The conclusion about MLL-AF4 translation inhibited by ribosomal stalling which occurs at AU-rich elements recognized by KHDRBS (in the abstract and discussion) is confusing as the authors show that IGF2BP3 is responsible for “the post-transcriptional inactivation of MLL-AF4”. While IGF2BPs are cytoplasmic, KHDRBS are located mostly in the nucleus. Functional screen for RBPs identifying *Igf2bp3* as an inhibitory factor of AF4 also raises a question about the role of other indicated proteins in regulation of

MLL-AF4 and their interplay, co-localization etc. The conclusion “MLL-AF4 is post-transcriptionally regulated, which explains why the establishment of a mouse disease model by MLL-AF4 has been difficult “is very general because literally all transcripts are bound to RBPs and are post-transcriptionally regulated. The manuscript would benefit from focusing on one protein with a proper characterization of RNA-protein interaction including the full-length mRNA transcripts.

Response: In this study, we demonstrated defective mutations of KHDRBS recognition sites in the *MLL-AF4* mRNA, and the *Igf2bp3* gene knock-out sufficiently de-activated the inhibitory regulation of protein translation of MLL-AF4. These results suggested that at least IGF2BP3 and KHDRBS are implicated in the regulation of MLL-AF4.

As reviewer #3 pointed out, IGF2BP3 and KHDRBS are localized differently in the cell, therefore expected to function separately. To examine whether these RBPs form an RNA/RBP complex on the same RNA, we performed a two-step pull-down assay (see Supplemental Fig. 6e). Our results indicate that KHDRBS and IGF2BP3 do not form a complex on the RNA simultaneously, suggesting that these proteins independently regulate the *MLL-AF4* mRNA in different steps of the mRNA maturation. We aim to elucidate the mechanism in more detail in the future. We have mentioned it in the discussion section (page 23 line 8).

As reviewer #3 mentioned, all transcripts are post-transcriptionally regulated. To avoid making a too general statement, we changed the title to “RNA-binding proteins of KHDRBS and IGF2BP families control the oncogenic activity of MLL-AF4”.

As reviewer #3 mentioned, the manuscript may benefit by focusing on one RBP instead of two. However, our results in this study demonstrated that both the mutations of KHDRBS recognition sites and knockout of *Igf2bp3* were required for the translation of MLL-AF4 proteins, which makes it reasonable to mention the two RBPs in this manuscript in our opinion.

As reviewer #3 pointed out, it would be good to perform analyses on the full-length mRNA transcripts. However, it is technically difficult at this point.

4. Identification of the cis-acting element responsible for induction of leukemia, even though it is not B-ALL, is a progress. However, the authors should clearly indicate what kind of leukemia was induced and what phenotypes they couldn't induce throughout the paper including abstract. RNA seq or ChIPseq would be a great addition. A cell type – mouse or human – used in manipulations should be mentioned more often through their manuscript including abstract. Information about spontaneous leukemia is irrelevant and should be removed from the manuscript.

Response: In our study, we were able to induce AML using MLL-AF4 synonymous mutants, but were unable to induce other phenotypes of leukemia in our condition. As reviewer #3 advised, we have mentioned the cell type (human or mouse) we used throughout our manuscript. We have removed the figures about spontaneous leukemia (Fig. S3c-i in the previous manuscript). As reviewer #3 suggested, we have performed an RNA-seq of bone marrow cells from AML mouse induced by a synonymous mutant of MLL-AF4 (Figure S3b, c in revised manuscript). The expression profiles of MLL-AF4 sPTRS-AMLs and

MLLmAf4-AMLs were similar to each other (Fig. S3b). MLL-fusion downstream genes such as *Hoxa9*, *Meis1*, and *Runx1*, were highly expressed in the AML cells, suggesting that the AML cells share the common phenotypes of leukemia by MLL fusions. The NGS data were deposited in Gene Expression Omnibus (GSE201503, token: abidyguydlmpzud).

Additional Notes:

1. RNA-binding proteins specifically associate with the PTRS of human AF4: you investigate protein binding with 210 or minimal 54bp fragment without a UV crosslink?

Response: Yes, we performed those pull-down analyses without UV crosslink .

2. Many RBPs (e.g., KHDRBS1/3, ELAVL1) specifically bound to hPTRSmin – in fact, many RBPs recognize similar or same primary and secondary RNA structures and compete for binding sites. Not only IGF2BPs can attenuate translation, stability etc.. The MLL-AF4 regulatory element you have identified is embedded in RNA.

Response: Yes, IGF2BPs are not the only ones that affect translation of AF4. We think that combinatorial functions by multiple RBPs (e.g., IGF2BPs, KHDRBSs) are involved. More detailed analysis to comprehensively dissect the networks of RBPs is required in the future.

References

1. Takahashi, S. *et al.* HBO1-MLL interaction promotes AF4/ENL/P-TEFb-mediated leukemogenesis. *Elife* **10** (2021).
2. Bursen, A. *et al.* The AF4.MLL fusion protein is capable of inducing ALL in mice without requirement of MLL.AF4. *Blood* **115**, 3570-3579 (2010).
3. Secker, K.A. *et al.* Only Hematopoietic Stem and Progenitor Cells from Cord Blood Are Susceptible to Malignant Transformation by MLL-AF4 Translocations. *Cancers (Basel)* **12** (2020).
4. Rice, S. *et al.* A human fetal liver-derived infant MLL-AF4 acute lymphoblastic leukemia model reveals a distinct fetal gene expression program. *Nat Commun* **12**, 6905 (2021).
5. Tran, T.M. *et al.* The RNA-binding protein IGF2BP3 is critical for MLL-AF4-mediated leukemogenesis. *Leukemia* **36**, 68-79 (2022).

Reviewers' Comments:

Reviewer #1:

Remarks to the Author:

This is a very interesting paper and a very well performed study. The results of the work are obviously a bit complicated, as it is still unclear why MLL-AF4 in human leukaemia cells is not inhibited by the same pathway, but I applaud the authors for their efforts. Overall I think that this paper contains important data and insights into MLL-AF4 biology that the field will find very useful.

Reviewer #2:

Remarks to the Author:

none

Reviewer #3:

Remarks to the Author:

The revised manuscript by Okuda et al. represents an essential improvement of their original paper.

1. The possible role of ncRNA in the regulation of MLL/AF4 transcript was investigated and discussed.

The RNA stability of that fusion was not significantly altered (Suppl. Fig.6b).

The quantification of WB images with GFP-hPTRS protein stability would be a great addition to strengthen the observation that there is less protein, to start with at "0", but it is not rapidly degraded (compared to GFP-WT, mPTRS, and sPTRS). The "kinetics of the degradation of the PTRS-fused reporter proteins", which is supposed to be in Supplementary Fig. 6c as well, was not found in that figure (?).

2. A significant body of evidence was provided to show that Igf2bp3 does not play a rigorous oncogenic role in MLLr leukemia. Even though this data remains somewhat controversial to some other reports, the post-transcriptional regulators can act as oncogenes or tumor suppressors depending on the context, concentration, and timing of their expression.

3. The two-step pull-down assay demonstrating that IGF2BP3 and KHDRBS do not co-localize with RNA is a good addition that illustrates the complexity of the regulatory mechanism of this fusion.

Mentioning both RBPs in the context of their subcellular localization and different regulatory roles is reasonable.

4. RNA-seq analysis of MLL-AF4sPTRS-AML and MLLmAF4 shows some similarities in the 2nd round of transplantation, but gene expression profiling is quite different at first. This is an interesting observation that indicates that MLL-hAF4 (yet sPTRS) and MLLmAF4 employ different mechanisms during transformation, which is also highlighted by your MLL-ENL-AML control.

5. Other comments and suggestions were addressed.

6. In the last paragraph, line 385, there is a typo - IGF3BP2

A point-by-point response to the reviewers' comments

REVIEWERS' COMMENTS

Reviewer #1 (Remarks to the Author):

This is a very interesting paper and a very well performed study. The results of the work are obviously a bit complicated, as it is still unclear why MLL-AF4 in human leukaemia cells is not inhibited by the same pathway, but I applaud the authors for their efforts. Overall I think that this paper contains important data and insights into MLL-AF4 biology that the field will find very useful.

Response: We appreciate the kind words and all of the comments and suggestions.

Reviewer #2 (Remarks to the Author):

none

Response: We appreciate all of the comments and suggestions during this review process.

Reviewer #3 (Remarks to the Author):

The revised manuscript by Okuda et al. represents an essential improvement of their original paper.

1. The possible role of ncRNA in the regulation of MLL/AF4 transcript was investigated and discussed.

The RNA stability of that fusion was not significantly altered (Suppl. Fig.6b).

Response: Yes.

The quantification of WB images with GFP-hPTRS protein stability would be a great addition to strengthen the observation that there is less protein, to start with at "0", but it is not rapidly degraded (compared to GFP-WT, mPTRS, and sPTRS). The "kinetics of the degradation of the PTRS-fused reporter proteins", which is supposed to be in Supplementary Fig. 6c as well, was not found in that figure (?).

Response: Western blotting data for the reporter proteins are shown in Supplementary Fig. 6c to demonstrate the kinetics.

2. A significant body of evidence was provided to show that Igf2bp3 does not play a rigorous oncogenic role in MLLr leukemia. Even though this data remains somewhat controversial to some other reports, the post-transcriptional regulators can act as oncogenes or tumor suppressors depending on the context, concentration, and timing of their expression.

Response: Yes, we discussed it in the discussion section.

3. The two-step pull-down assay demonstrating that IGF2BP3 and KHDRBS do not co-localize with RNA is a good addition that illustrates the complexity of the regulatory mechanism of this fusion.

Response: We appreciate the feedback.

Mentioning both RBPs in the context of their subcellular localization and different regulatory roles is reasonable.

Response: We appreciate the feedback.

4. RNA-seq analysis of MLL-AF4sPTRS-AML and MLLmAF4 shows some similarities in the 2nd round of transplantation, but gene expression profiling is quite different at first. This is an interesting observation that indicates that MLL-hAF4 (yet sPTRS) and MLLmAF4 employ different mechanisms during transformation, which is also highlighted by your MLL-ENL-AML control.

Response: It is indeed interesting. The expression of MLL-AF4sPTRS-AML and MLLmAF4 may require slightly different cell-of-origin and go through some adaptation process during the evolution of leukemia.

5. Other comments and suggestions were addressed.

Response: We appreciate all of the comments and suggestions.

6. In the last paragraph, line 385, there is a typo - IGF3BP2

Response: We have corrected the typo.